# Efficient Learning of Linear Graph Neural Networks via Node Subsampling

**Seiyun Shin[1], Ilan Shomorony[1], and Han Zhao[2]**
[1]Department of ECE and [2]Department of CS
University of Illinois at Urbana-Champaign, IL
{seiyuns2, ilans, hanzhao}@illinois.edu

## Abstract

Graph Neural Networks (GNNs) are a powerful class of machine learning models with applications in recommender systems, drug discovery, social network analysis, and computer vision. One challenge with their implementation is that GNNs often take large-scale graphs as inputs, which imposes significant computational/storage costs in the training and testing phases. In particular, the message passing operations of a GNN require multiplication of the graph adjacency matrix $A \in \mathbb{R}^{n \times n}$ and the data matrix $X \in \mathbb{R}^{n \times d}$, and the $O(n^2 d)$ time complexity can be prohibitive for large $n$. Thus, a natural question is whether it is possible to perform the GNN operations in (quasi-)linear time by avoiding the full computation of $AX$. To study this question, we consider the setting of a regression task on a two-layer Linear Graph Convolutional Network (GCN). We develop an efficient training algorithm based on (1) performing node subsampling, (2) estimating the leverage scores of $AX$ based on the subsampled graph, and (3) performing leverage score sampling on $AX$. We show that our proposed scheme learns the regression model observing only $O(nd\varepsilon^{-2} \log n)$ entries of $A$ in time $O(nd^2\varepsilon^{-2} \log n)$, with the guarantee that the learned weights deviate by at most $\varepsilon$ under the $\ell_2$ norm from the model learned using the entire adjacency matrix $A$. We present empirical results for regression problems on real-world graphs and show that our algorithm significantly outperforms other baseline sampling strategies that exploit the same number of observations.

## 1   Introduction

Graph Neural Networks (GNNs) have gained popularity as a powerful machine learning method for graph-structured data. By learning rich representations of graph data, GNNs can solve a variety of prediction tasks on graphs [1–7]. GNNs have delivered impressive results across many different areas, including social network analysis [8], bioinformatics [9–11], and recommendation systems [12].

Given their remarkable performance, being able to train GNNs efficiently is an important task. However, training GNNs can be quite challenging in the context of large-scale graphs, which impose significant computational costs. In particular, the message passing scheme in GNNs requires, for each node, summing up the feature vectors of all neighboring nodes into one feature vector. For a multi-layer Graph Convolutional Network (GCN) [3], the layer-wise propagation rule is

$$H^{(\ell+1)} = \sigma(A H^{(\ell)} W^{(\ell)}), \tag{1}$$

where $H^{(\ell)}$ denotes the feature representation at the $\ell$th layer, with $H^{(0)} = X$ (i.e., the data matrix consisting of nodes' features of an input graph $G$); $W^{(\ell)}$ denotes the weight matrix; $\sigma(\cdot)$ is a non-linear activation function like the ReLU; and $A$ denotes the adjacency matrix of $G$.

37th Conference on Neural Information Processing Systems (NeurIPS 2023).

If the graph has $n$ nodes and the feature dimension is $d$, computing the matrix multiplication $AX$ requires $O(n^2 d)$ time and can be prohibitive in big data settings. A natural question is whether it is possible to train a GCN to avoid the quadratic complexity scaling with $n$.

Another motivation for avoiding the full computation of $AX$ is applications where the adjacency matrix $A$ is not fully known a priori, and it must be learned via node/edge queries. For example, in large social networks (the current number of social media users is over $4.89$ billion [13]), one may need to access the adjacency matrix $A$ by querying the adjacency list of specific nodes. As another example, certain biological networks must be learned through the physical probing of pairwise interactions, which may make obtaining the entire adjacency matrix $A$ prohibitively expensive. For example, the mapping of the neural network of living organisms (such as the connectome of *C. elegans* [14]) requires physical probing of the connectivity between neurons.

A natural approach to avoid the quadratic complexity of computing $AX$ is via the subsampling of the graph $G$ [15, 6, 16, 17]. If the resulting subgraph is sparse enough, $AX$ can be computed very efficiently. But how many entries of $A$ need to be observed in order to guarantee that the message passing step $AX$ can be computed accurately enough in the context of GNNs? What kinds of graph subsampling strategies are amenable to theoretical performance guarantees on the training of GNNs?

We consider the setting of a regression task to be learned via a GCN. Let $A \in \mathbb{R}^{n \times n}$ be the weighted adjacency matrix of a graph $G = (V, A)$ with $|V| = n$. Each node $v_i \in V$ has an associated feature vector of dimension $d$ $(< n)$ and a label, denoted by $(\mathbf{x}_i, y_i) \in \mathbb{R}^d \times \mathbb{R}$, which can also be represented as a $n \times d$ data matrix $X$ and label vector $\mathbf{y}$. Training the GCN corresponds to minimizing the loss

$$\mathcal{L}(W, A, X, \mathbf{y}) := \tfrac{1}{2n} \|\mathbf{y} - f_{\text{GCN}}(W, A, X)\|_2^2, \tag{2}$$

on the training data $\{(\mathbf{x}_i, y_i)\}_{i=1}^n$, where $W$ denotes the GCN network weights, and $f_{\text{GCN}}(W, A, X)$ denotes a feed-forward computation of a GCN's output. As a first step to studying whether (2) can be solved accurately on a sparse subgraph of $G$, we focus on a simple *linear* GCN, where there is no non-linearity $\sigma(\cdot)$. Specifically, the feed-forward output is given by $f_{\text{GCN}}^{\text{linear}}(\mathbf{w}, A, X) := AX\mathbf{w}$, where we use $\mathbf{w} \in \mathbb{R}^d$ (instead of $W \in \mathbb{R}^{1 \times d}$) to indicate that the learnable parameters are in the form of a vector. Hence, our goal is to solve

$$\min_{\mathbf{w}} \quad \tfrac{1}{2n} \|\mathbf{y} - AX\mathbf{w}\|_2^2. \tag{3}$$

Note that one can view this optimization problem as a *graph-weighted linear regression problem*.

The setting of linear regression provides natural suggestions for a subsampling strategy. It is known that *leverage score sampling* allows one to solve the linear regression problem with $(1 + \varepsilon)$ accuracy using only $O(d\varepsilon^{-2} \log n)$ subsampled rows of the data matrix $X$ and in time $O(nd^2 + d^3)$[1] [18–22]. Nevertheless, to apply this strategy to (3), one would need to first compute $AX$, requiring $O(n^2 d)$ time, so that the row leverage scores of $AX$ can then be computed. This motivates us to propose a two-step approach that (1) performs node subsampling to obtain an approximate observation of $AX$ and estimate the leverage scores of the rows of the augmented matrix $[AX| - \mathbf{y}] \in \mathbb{R}^{n \times (d+1)}$ and (2) performs leverage score sampling on the rows of $[AX| - \mathbf{y}]$ using the estimated scores.

This two-step approach is illustrated in Figure 1. In order to obtain an estimator of $AX$, our sampling scheme in the first step builds $O(\log n)$ rank-1 matrices by sampling the same amount of columns of $A$ and the corresponding rows of $X$. The key idea behind this approach is taking a random subset of nodes and propagating their feature vectors to their neighbors. We provide a spectral approximation guarantee for the resulting estimate of $AX$ and an approximation guarantee for the leverage scores of the augmented matrix $[\widehat{AX}| - \mathbf{y}]$ computed on this estimate. In the second step, we adopt the standard leverage score sampling for sampling rows of $A$, using the leverage score estimates of $[AX| - \mathbf{y}]$ obtained from the first step. With $O(d\varepsilon^{-2} \log n)$ sampled rows of $\tilde{A}$ and $\tilde{y}$, the training algorithm then computes $\tilde{A}X$ and uses it to solve the regression problem. We show that our proposed scheme learns the regression model with learned weights deviated by at most $\varepsilon$ from those with full information, by observing only $O(nd\varepsilon^{-2} \log n)$ entries of $A$, and in time $O(nd^2\varepsilon^{-2} \log n)$.

On real-world benchmark datasets, we demonstrate the performance improvements of the two proposed schemes over other baseline sampling schemes via numerical simulations.

---

[1]More precisely, the time complexity is $\tilde{O}((\text{nnz}(AX) + d^3) \log(\varepsilon^{-1}))$, where $\tilde{O}(\cdot)$ hides factors polylogarithmic in $(n, d)$.

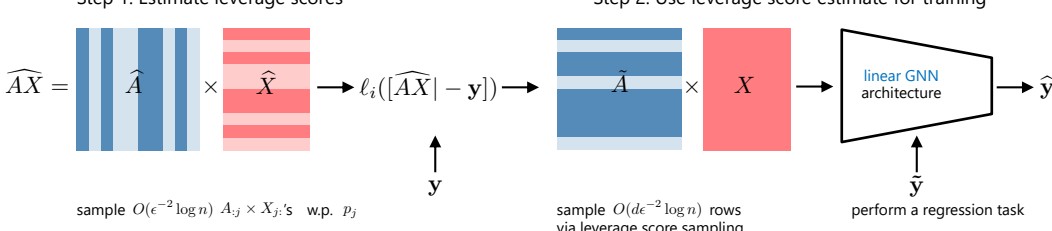

Step 1: Estimate leverage scores                    Step 2: Use leverage score estimate for training

sample $O(\epsilon^{-2}\log n)$ $A_{:j} \times X_{j:}$'s w.p. $p_j$        sample $O(d\epsilon^{-2}\log n)$ rows        perform a regression task
                                                                          via leverage score sampling

Figure 1: Two-step Algorithm: (1) perform node subsampling to obtain an estimate $\widehat{AX}$, from which we can compute $\ell_i([\widehat{AX})| - \mathbf{y}]$ for $i = 1, \dots, n$; and (2) use the leverage score estimates to perform leverage score sampling on $A$ and $\mathbf{y}$ and perform regression using $\tilde{A}X$ and $\tilde{\mathbf{y}}$ instead of $AX$ and $\mathbf{y}$.

**Notation:** Throughout the paper, we use $A = (a_{ij}) \in \mathbb{R}_{\geq 0}^{n \times n}$ to denote the graph adjacency matrix, where each entry value is nonnegative and bounded (i.e., $0 \leq a_{ij} \leq M$ for some constant $M > 0$). We write $A_{i:}$ and $A_{:j}$ to indicate the $i$th row and $j$th column of the matrix $A$ respectively. In addition, $X \in \mathbb{R}^{n \times d}$ denotes the data (design) matrix, where the $i$th row in $X$, $\mathbf{x}_i$, corresponds to the node vector at the $i$th node $v_i$ in the graph $G$. We also assume that the absolute value of each entry value of $X$ is bounded by $M$ (i.e., $|X_{ij}| \leq M$). We denote vectors by a lowercase bold letter (e.g., $\mathbf{x} \in \mathbb{R}^n$), and by default, all the vectors will be in column format, hence $X = (\mathbf{x}_1^\top; \dots; \mathbf{x}_n^\top)^\top$, where $(\cdot)^\top$ denotes the transpose. We indicate the $i$th entry of $\mathbf{x}$ with either $[\mathbf{x}]_i$ or $x_i$, and we use $[\mathbf{x}]_S$ to denote the concatenation of $x_i$s for all $i \in S$. We also let $[n] := \{1, 2, \dots, n\}$. We denote by $\langle \mathbf{x}, \mathbf{y} \rangle := \sum_i x_i y_i \in \mathbb{R}$ the inner product between $\mathbf{x}$ and $\mathbf{y}$. Unless otherwise mentioned, we use $\|\mathbf{v}\|$ for the $\ell_2$-norm for vector $\mathbf{v}$. We let $\|A\| := \max_{\|\mathbf{v}\|=1} \|A\mathbf{v}\|$ denote the operator norm of a matrix, and $\|A\|_F := \sqrt{\sum_{i,j} a_{ij}^2}$ denote the Frobenius norm. We denote the standard basis vector by $\mathbf{e}_i$. We write $\mathrm{Bern}(p)$ for a Bernoulli distribution with parameter $p$. Lastly, we denote by $\mathrm{nnz}(A)$ the number of non-zero entries in matrix $A$.

**Related Work:** We provide a detailed discussion of related works in Appendix A.

## 2 Motivation: Leverage Score Sampling for Linear Regression

First, we briefly describe key results on leverage score sampling that will be relevant to our discussion. We refer to [23] for a detailed discussion on the subject, and we provide key proofs in Appendix D.

**Definition 1.** The leverage score of the $i$th row of $X$ is $\ell_i(X) := \mathbf{x}_i^\top (X^\top X)^\dagger \mathbf{x}_i$, where $(\cdot)^\dagger$ denotes the Moore-Penrose pseudoinverse.

Intuitively, a row's leverage score measures how important it is in composing the row space of $X$. If a row has a component orthogonal to all the other rows, its leverage score is 1. Removing it would decrease the rank of $X$, completely changing its row space. The leverage scores are the diagonal entries of the projection matrix $X(X^\top X)^{-1}X^\top$, which can be used to show that $\sum_{i=1}^n \ell_i(X) = \mathrm{rank}(X)$. The following alternative characterizations of $\ell_i(X)$ will be useful.

**Proposition 1.** Let $X = U\Sigma V^\top$ be the singular value decomposition (SVD) of $X$, where $U \in \mathbb{R}^{n \times d}$, $\Sigma \in \mathbb{R}^{d \times d}$ and $V \in \mathbb{R}^{d \times d}$. Then $\ell_i(X) = \|U_{i:}\|_2^2$, where $U_{i:}$ is the $i$th row of $U$.

**Proposition 2.** The leverage score can be alternatively computed as $\ell_i(X) = \max_{\mathbf{v}} \frac{(\mathbf{x}_i^\top \mathbf{v})^2}{\|X\mathbf{v}\|_2^2}$.

Through the use of leverage scores, it is possible to approximately solve a linear regression task using a subset of the data points [18–22]. In particular, leverage score sampling allows the spectral approximation of a matrix.

**Definition 2.** A matrix $\widehat{X} \in \mathbb{R}^{n \times d}$ is an $\varepsilon$-spectral approximation of $X \in \mathbb{R}^{n \times d}$ if, for all $\mathbf{v} \in \mathbb{R}^d$,

$$(1 - \varepsilon) \cdot \|X\mathbf{v}\|_2 \leq \|\widehat{X}\mathbf{v}\|_2 \leq (1 + \varepsilon) \cdot \|X\mathbf{v}\|_2. \tag{4}$$

Suppose we are given a data matrix $X = (\mathbf{x}_1^\top; \dots; \mathbf{x}_n^\top)^\top$ and a vector $\mathbf{u} = (u_1, \dots, u_n)$ of leverage score *overestimates* of $X$ (i.e., $\ell_i(X) \leq u_i, \forall i \in [n]$). We create a matrix $\tilde{X}$ by including the $i$th row

of $X$ in $\tilde{X}$ with probability $p_i = \min[1, cu_i\varepsilon^{-2}\log n]$ for some constant $c$, scaled by $1/\sqrt{p_i}$. The following is a slight modification of Lemma 4 in [21] (which we prove in Appendix D):

**Lemma 1.** With probability at least $1 - n^{-\Omega(1)}$, for any $\mathbf{v}$, we have

$$(1-\varepsilon)\cdot\|X\mathbf{v}\| \le \|\tilde{X}\mathbf{v}\| \le (1+\varepsilon)\cdot\|X\mathbf{v}\|.$$

Notice that the expected number of rows in $\tilde{X}$ is $c\|\mathbf{u}\|_1\varepsilon^{-2}\log n$. Hence, as long as $\|\mathbf{u}\|_1 = O(d)$, we only need to sample $O(d\varepsilon^{-2}\log n)$ rows of $X$ to obtain a good spectral approximation.

Now suppose we want to solve the OLS problem $\min_{\mathbf{w}}\frac{1}{2}\|\mathbf{y} - X\mathbf{w}\|_2^2$ where $\mathbf{y} \in \mathbb{R}^n$ is the response vector corresponding to $X$. The optimal solution is given by $\mathbf{w}^* = (X^\top X)^{-1}X^\top\mathbf{y}$. Exactly computing $\mathbf{w}^*$ requires $O(nd^2 + d^3)$ time.

An alternative approach using leverage score sampling would be as follows. First, we note that the OLS objective can be rewritten using $\|\mathbf{y} - X\mathbf{w}\| = \|[X|-\mathbf{y}]\cdot[{}^{\mathbf{w}}_1]\|$. If we now have leverage score overestimates $u_1, \ldots, u_n$ for the augmented matrix $[X|-\mathbf{y}] \in \mathbb{R}^{n\times(d+1)}$, we can construct the subsampled matrix $[X_S|-\mathbf{y}_S] \in \mathbb{R}^{|S|\times(d+1)}$ (with rows rescaled by $1/\sqrt{p_i}$). Then, by Lemma 1, a solution to

$$\tilde{\mathbf{w}} = \arg\min_{\mathbf{w}}\quad\left\|[X_S|-\mathbf{y}_S]\cdot\begin{bmatrix}\mathbf{w}\\1\end{bmatrix}\right\|_2,\tag{5}$$

would satisfy $\|\mathbf{y}_S - X_S\tilde{\mathbf{w}}\|_2 \le \|\mathbf{y}_S - X_S\mathbf{w}^*\|_2 \le (1+\varepsilon)\|\mathbf{y} - X\mathbf{w}^*\|_2$.

The exact leverage scores of $X$ can be computed in time $O(nd^2)$ (see Algorithm 3 in Appendix B) and, since $|S| = O(d\varepsilon^{-2}\log n)$, the reduced linear system (5) can be solved in time $O(d^3\varepsilon^{-2}\log n)$. The overall complexity for solving the OLS problem in this way is given by $O(nd^2 + d^3\varepsilon^{-2}\log n)$. Notice that there is no real acceleration obtained by this procedure. But also notice that here we are using the exact leverage scores for sampling while, from Lemma 1, we know that this approach works as long as we have good leverage score overestimates.

## 3 Efficient Training of a Linear GCN via Leverage Scores

In this section, we introduce an efficient training algorithm that approximately solves the graph-weighted regression task (3) by observing only $O(nd\varepsilon^{-2}\log n)$ entries of $A$, and in time $O(nd^2\varepsilon^{-2}\log n)$. The necessity for introducing the efficient training algorithm comes from the $O(n^2d)$ time complexity of computing $AX$ via matrix multiplication.

Motivated by the discussion in Section 2 for standard linear regression, a natural approach is to attempt to compute leverage score overestimates for $AX$ efficiently without the full computation of $AX$. Our proposed algorithm uses the following three steps to approximately solve the graph-weighted linear regression problem.

1. ESTIMATELEVERAGESCORES$(A, X)$ (See Algorithm 1 and Algorithm 2)

2. LEVERAGESCORESAMPLING$(A, \{\widehat{\ell}_i([AX|-\mathbf{y}]),\ \forall i \in [n]\})$ (See Algorithm 4 in Appendix B)

3. REGRESSIONSOLVER$(\tilde{A}X, \tilde{\mathbf{y}})$ (See Algorithm 5 in Appendix B)

In ESTIMATELEVERAGESCORES$(A, X)$, a node subsampling technique is used to build an estimate $\widehat{AX}$ of $AX$ with an approximation guarantee. The estimate $\widehat{AX}$ can then be used to produce provably good overestimates of the leverage scores of the augmented matrix $[AX|-\mathbf{y}]$. Next, LEVERAGESCORESAMPLING$(A, \{\widehat{\ell}_i([AX|-\mathbf{y}]),\ \forall i \in [n]\})$ uses these overestimates to produce a matrix $\tilde{A}$ and scaled labels $\tilde{\mathbf{y}}$ consisting of a reduced number of rows of $A$ and $\mathbf{y}$ respectively. Using Lemma 1, we then show that $\tilde{A}X$ provides a spectral approximation for $AX$, which allows approximately solving the regression problem (3) using REGRESSIONSOLVER$(\tilde{A}X, \tilde{\mathbf{y}})$. We note that none of the algorithmic procedures requires fully computing $AX$ and the end-to-end algorithm has an $O(n\log n)$ time dependence on $n$.

---
**Algorithm 1** ESTIMATELEVERAGESCORES$(A, X)$ via Uniform Sampling
---
**Input:** Adjacency matrix $A \in \mathbb{R}^{n \times n}$, data matrix $X \in \mathbb{R}^{n \times d}$, budget $B$, threshold $\varepsilon > 0$
**Output:** Leverage score overestimates for $AX$

1: Draw $I_j \sim \text{Bern}(p)$, $\forall j \in [n]$ independently, where $p = \min\left[\frac{B}{n}, 1\right]$
2: $\widehat{A_{:j}X_{j:}} \leftarrow \frac{I_j}{p} A_{:j} X_{j:}$, $\forall j \in [n]$
3: $\widehat{AX} \leftarrow \sum_{j=1}^{n} \widehat{A_{:j}X_{j:}}$                           $\triangleright O\left(Bd\right)$
4: $S \leftarrow$ LEVERAGESCORE$([\widehat{AX}| - \mathbf{y}])$        $\triangleright$ See Alg. 3 in App. B; $\tilde{O}(\text{nnz}([\widehat{AX}| - \mathbf{y}]) + d^3)$
5: $\{\widehat{\ell}_i([AX| - \mathbf{y}]), \forall i \in [n]\} \leftarrow \frac{(1+\varepsilon)^2}{(1-\varepsilon)^2} \cdot S$
6: **return** $\widehat{\ell}_i([AX| - \mathbf{y}]), \forall i \in [n]$
---

## 3.1 First Stage: Uniform Sampling of $A_{:j}X_{j:}$

In order to efficiently estimate the leverage scores of $AX$, a key observation we make is that $AX$ can be decomposed as a sum of $n$ rank-1 matrices as $AX = \sum_{j=1}^{n} A_{:j}X_{j:}$, where $A_{:j}$ and $X_{j:}$ denote $j$th column of the adjacency matrix $A$ and $j$th row of the data matrix $X$. Notice that one can view $A_{:j}X_{j:}$ as the effect of node $j$'s feature vector propagated to other nodes after the one-step message passing operation. Hence, a natural approximation strategy is to take a subset of these rank-1 matrices. First, we consider a uniform sampling strategy that samples node indices $j$ uniformly at random and computes $A_{:j}X_{j:}$. See Algorithm 1 for the detailed procedure.

Given a budget $B$ (the total number of nodes that should be observed), the algorithm draws sampling indicator variables $I_j \sim \text{Bern}(p)$ independently with $p = \min[B/n, 1]$. For $j = 1, \ldots, n$, we build

$$\widehat{A_{:j}X_{j:}} = \frac{I_j}{p} A_{:j} X_{j:},$$

and then set $\widehat{AX} := \sum_j \widehat{A_{:j}X_{j:}}$. Notice that $\widehat{AX}$ is an unbiased estimate of $AX$, because for all $i, j$,

$$\mathbb{E}\left[[\widehat{AX}]_{ij}\right] = \sum_{k=1}^{n} p \cdot \frac{1}{p} a_{ik} x_{kj} = \sum_{k=1}^{n} a_{ik} x_{kj} = [AX]_{ij}. \tag{6}$$

In addition, one can readily see that $B$ rank-1 matrices are sampled in expectation, and, with high probability, the actual number of sampled rank-1 matrices is less than $(1 + \alpha)B$ for $\alpha > 0$.

Next, we present a spectral approximation guarantee for $\widehat{AX}$. In order to establish that result, we make the following mild assumption on the density of the input graph.

*Assumption* 1. We assume that the input graph $G$ satisfies

$$\|AX\| \geq n^{3/2} d^{1/2}. \tag{7}$$

Strictly speaking, (7) is not just an assumption about the density of $G$, as it takes into consideration the interaction between $A$ and $X$. Nevertheless, this assumption would be expected to hold in cases where the graph $G$ is dense. In particular, since $\|AX\| \geq d^{-1/2}\|AX\|_F$, Assumption 1 holds if $\|AX\|_F \geq n^{3/2}d$. Assuming that $A$ has $\delta n^2$ non-zero entries and that entries of $X$ are drawn i.i.d. from some distribution with positive second moment $\beta^2$, independently of $A$, we have $\|AX\|_F \approx \sqrt{\delta n^2 d \beta^2} = \beta \delta^{1/2} n^{3/2} d^{1/2}$. Treating $d$ as a constant and rescaling $X$ by a constant, we have that the assumption holds. More generally, if $d$ is a constant and

$$\left|\{(i, j) \in [n] \times [d] : \mathbf{e}_i^\top AX\mathbf{e}_j \geq c_1 n\}\right| \geq c_2 n$$

for constants $c_1$ and $c_2$, then the assumption holds (after a constant rescaling of the data matrix). The following stronger assumption will be needed to establish guarantees for the leverage score estimates:

*Assumption* 2. For all $(i, j) \in [n] \times [d]$, $\mathbf{e}_i^\top AX\mathbf{e}_j \geq c_1 n$ for some constant $c_1$.

Notice that if the graph $G$ is instead sparse, the problem we focus on due to the computation of $AX$ is less severe since one can efficiently compute $AX$ in time that depends on $\text{nnz}(A)$.

The proof of the following theorem is in Appendix E.

**Theorem 1.** Under Assumption 1, if we run Algorithm 1 with budget $B = C\varepsilon^{-2} \log n$ for large enough $C > 0$, the estimate $\widehat{AX}$ satisfies

$$(1 - \varepsilon) \cdot \|AX\mathbf{v}\| \le \|\widehat{AX}\mathbf{v}\| \le (1 + \varepsilon) \cdot \|AX\mathbf{v}\|, \quad \forall \mathbf{v} \in \mathbb{R}^d, \tag{8}$$

with probability at least $1 - n^{-\Omega(1)}$.

Given the spectral approximation guarantee in Theorem 1, we expect the leverage scores computed on the augmented matrix $[\widehat{AX}| - \mathbf{y}]$ to be close to the leverage scores of $[AX| - \mathbf{y}]$. Since we want an overestimate of the leverage scores, we use the the multiplicative constant $(\frac{1+\varepsilon}{1-\varepsilon})^2$ to obtain

$$\widehat{\ell}_i([AX| - \mathbf{y}]) := \left(\frac{1+\varepsilon}{1-\varepsilon}\right)^2 \cdot \ell_i([\widehat{AX}| - \mathbf{y}]), \quad \forall i \in [n]. \tag{9}$$

The following lemma, whose proof is in Appendix F, asserts that $\widehat{\ell}_i([AX| - \mathbf{y}])$ is indeed an overestimate of $\ell_i([AX| - \mathbf{y}])$.

**Lemma 2.** Under Assumption 2, Algorithm 1 with budget $B = C\varepsilon^{-2} \log n$ for large enough $C > 0$ outputs leverage score overestimates satisfying

$$\ell_i([AX| - \mathbf{y}]) \le \widehat{\ell}_i([AX| - \mathbf{y}]) \le \left(\frac{1+\varepsilon}{1-\varepsilon}\right)^4 \ell_i([AX| - \mathbf{y}]), \quad \forall i \in [n], \tag{10}$$

with probability at least $1 - n^{-\Omega(1)}$.

### 3.2 Second Stage: Leverage Score Sampling and Approximate Regression

The second step of the proposed algorithm exploits the leverage score estimates $\widehat{\ell}_i([AX| - \mathbf{y}])$, $\forall i \in [n]$ to perform the standard leverage score sampling. With Lemma 1, one can readily obtain a $(1 + \varepsilon)$-approximation guarantee for the graph-weighted linear regression task.

**Theorem 2.** Under Assumption 2, suppose we run Algorithm 1 with budget $B = C\varepsilon^{-2} \log n$ for large enough $C > 0$, to obtain $\{\widehat{\ell}_i([AX| - \mathbf{y}]), \forall i \in [n]\}$. Subsequently, if we run LEVERAGESCORESAMPLING$(A, \{\widehat{\ell}_i([AX| - \mathbf{y}]), \forall i \in [n]\})$, we obtain $\tilde{A} \in \mathbb{R}^{O(d\varepsilon^{-2} \log n) \times n}$ and $\tilde{\mathbf{y}} \in \mathbb{R}^{O(d\varepsilon^{-2} \log n)}$ such that REGRESSIONSOLVER$(\tilde{A}X, \tilde{\mathbf{y}})$ provides $\tilde{\mathbf{w}}$ satisfying

$$\|\tilde{\mathbf{y}} - \tilde{A}X\tilde{\mathbf{w}}\|_2^2 \le (1 + \varepsilon) \cdot \min_{\mathbf{w}} \|\mathbf{y} - AX\mathbf{w}\|_2^2.$$

Theorem 2 captures our main contribution. By only observing $O(d \log n)$ rows of $A$, we can obtain a reduced adjacency matrix $\tilde{A}$ (and corresponding label vector $\tilde{\mathbf{y}}$). Running REGRESSIONSOLVER$(\tilde{A}X, \tilde{\mathbf{y}})$ then yields a solution $\tilde{\mathbf{w}}$ with an approximation guarantee. We refer to detailed descriptions of LEVERAGESCORESAMPLING$(A, \{\widehat{\ell}_i([AX| - \mathbf{y}]), \forall i \in [n]\})$ and REGRESSIONSOLVER$(\tilde{A}X, \tilde{\mathbf{y}})$ in Appendix B.

**Sample complexity and run-time:** We note that the sample complexity is $O(nd\varepsilon^{-2} \log n)$. First, ESTIMATELEVERAGESCORES$(A, X)$ exploits $O(\varepsilon^{-2} \log n)$ columns of $A$ and the same number of rows of $X$, thus accessing $O(n\varepsilon^{-2} \log n)$ entries of $A$ and $O(d\varepsilon^{-2} \log n)$ of $X$. In addition, LEVERAGESCORESAMPLING$(A, \{\widehat{\ell}_i([AX| - \mathbf{y}]), \forall i \in [n]\})$ exploits $O(d\varepsilon^{-2} \log n)$ rows and thus uses $O(nd\varepsilon^{-2} \log n)$ entries. Hence, the total number of entry observations is $O(nd\varepsilon^{-2} \log n)$.

For the time complexity, the run time of ESTIMATELEVERAGESCORES$(A, X)$ is $O(nd\varepsilon^{-2} \log n)$; that of LEVERAGESCORESAMPLING$(A, \{\widehat{\ell}_i([AX| - \mathbf{y}]), \forall i \in [n]\})$ is $O(nd^2\varepsilon^{-2} \log n)$. Finally, we note that the run time of REGRESSIONSOLVER$(\tilde{A}X, \tilde{\mathbf{y}})$ is $O(nd^2 + d^3)^2$. Hence, the total run time is $O(nd^2\varepsilon^{-2} \log n)$ given the setting that we consider has $d < n$.

---

[2]We refer to [24–28] that introduce a variety of acceleration algorithms for regression.

**Extension to the more-than-two-layer case:** Note that without non-linearities, considering a multi-layer network is equivalent to performing additional matrix multiplication by $A$ to the output of the previous layer. Since our proposed scheme yields a spectral approximation guarantee for matrix multiplication, an analysis similar to that of Algorithm 1 should be possible for this extension to obtain a $(1 + \varepsilon)^L$-approximation guarantee, where $L$ denotes the number of layers. Since the approximation guarantee comes with high probability, a union bound could be used to obtain a high probability guarantee for the multi-layer case (since the order or error for each approximate matrix multiplication is the same).

## 4 Variance Minimization via Data-dependent Random Sampling

As stated in the previous section, Algorithm 1 produces an unbiased estimate of $AX$. Yet the uniform sampling strategy used in Algorithm 1 does not take the variance of the estimator into consideration. This motivates us to develop an estimator of $AX$ that further reduces the estimator variance while maintaining unbiasedness. Due to the unbiasedness of $\widehat{AX}$, the total variance of the estimator coincides with the mean square error (MSE): $\mathrm{Var}(\widehat{AX}) = \mathbb{E}\|AX - \widehat{AX}\|_F^2 = \mathrm{MSE}(\widehat{AX})$.

In order to reduce the estimator variance $\mathrm{Var}(\widehat{AX})$, we consider a data-dependent sampling probability $p_j$ that depends on the $j$th column of $A$ and the $j$th row of $X$. As we view $A_{:j}X_{j:}$ as capturing the impact of node $j$'s features after a one-step message passing operation, one way to set the sampling probability for node $j$ is to take the norm of $A_{:j}X_{j:}$ into account. Specifically, we set

$$p_j = \min\left[B \cdot \frac{\|A_{:j}\|\|X_{j:}\|}{\sum_j \|A_{:j}\|\|X_{j:}\|}, 1\right]. \tag{11}$$

Algorithm 2 describes the detailed procedure, where the only distinction relative to Algorithm 1 is the choice of $p_j$. We claim that this sampling probability choice minimizes the total variance (as well as the MSE) of the estimator. First notice that the total variance can be expressed as

$$\mathrm{Var}(\widehat{AX}) = \sum_{i=1}^n \sum_{j=1}^d \mathrm{Var}\left([\widehat{AX}]_{ij}\right) \overset{(i)}{=} \sum_{k=1}^n \left(\frac{1}{p_k}\left(\sum_{i=1}^n a_{ik}^2\right)\left(\sum_{j=1}^d x_{kj}^2\right)\right) - C_2 \ (C_2 := ndC_1)$$

$$\overset{(ii)}{=} \left(\sum_{k=1}^n \frac{\|A_{:k}\|^2\|X_{k:}\|^2}{p_k}\right) \cdot \left(\sum_{k=1}^n p_k\right) - C_2$$

$$\overset{(iii)}{\geq} \left(\sum_{k=1}^n \|A_{:k}\|\|X_{k:}\|\right)^2 - C_2,$$

where $(i)$ follows from the fact that

$$\mathrm{Var}\left([\widehat{AX}]_{ij}\right) = \sum_{k=1}^n \mathrm{Var}\left(\frac{I_k}{p_k}a_{ik}x_{kj}\right) = \sum_{k=1}^n \frac{1}{p_k^2}a_{ik}^2 x_{kj}^2 \mathrm{Var}\left(I_k\right) = \sum_{k=1}^n \frac{1 - p_k}{p_k}a_{ik}^2 x_{kj}^2$$

$$= \sum_{k=1}^n \frac{1}{p_k}a_{ik}^2 x_{kj}^2 - \sum_{k=1}^n a_{ik}^2 x_{kj}^2 = \sum_{k=1}^n \frac{1}{p_k}a_{ik}^2 x_{kj}^2 - C_1 \ \text{(for some constant } C_1)$$

and $(ii)$ follows from using the fact that $\sum_j p_j = 1$; and $(iii)$ follows from Cauchy-Schwarz inequality. Notice that, in order for Cauchy-Schwarz to hold with equality, we must have $p_k$ proportional to $\|A_{:k}\|\|X_{k:}\|$. This implies that (11) is an optimal choice of the sampling probabilities.

As it turns out, the same theoretical guarantees obtained for Algorithm 1 hold in this case. See Appendix G for the detailed proof. While the theoretical performance guarantee is the same, in settings where the budget $B$ is very small, the data-dependent sampling probabilities $p_j$ lead to a better estimate $\widehat{AX}$, which improves the overall performance of the algorithm.

One important observation is that it is not obvious how to use the data-dependent sampling probabilities $p_j$ while avoiding the an $O(n^2)$ computation complexity. In particular, a naive computation of $\|A_{j:}\|$, $j \in [n]$ would require observing all $n^2$ entries of $A$. However, standard sampling techniques could be used in order to also estimate $\|A_{j:}\|$, $j \in [n]$ fairly accurate, and use those estimates to compute the sampling probabilities $p_j$, $j \in [n]$.

---

**Algorithm 2** ESTIMATELEVERAGESCORES$(A, X)$ via Data-dependent Sampling

---

**Input:** Adjacency matrix $A \in \mathbb{R}^{n \times n}$, data matrix $X \in \mathbb{R}^{n \times d}$, budget $B$, threshold $\varepsilon > 0$
**Output:** Leverage score estimates $\widehat{\ell}_i([AX| - \mathbf{y}]), \forall i \in [n]$

---

1: Draw $I_j \sim \text{Bern}(p_j), \forall j \in [n]$ independently, where $p_j = \min \left[ B \cdot \frac{\|A_{:j}\| \|X_{j:}\|}{\sum_j \|A_{:j}\| \|X_{j:}\|}, 1 \right]$
2: $\widehat{A_{:j} X_{j:}} \leftarrow \frac{I_j}{p_j} A_{:j} X_{j:}, \forall j \in [n]$
3: $\widehat{AX} \leftarrow \sum_{j=1}^{n} \widehat{A_{:j} X_{j:}}$  $\qquad\qquad\qquad\qquad\qquad\qquad$ ▷ $O(Bd)$
4: $S \leftarrow$ LEVERAGESCORE$([\widehat{AX}| - \mathbf{y}])$ $\qquad\qquad\qquad$ ▷ $\tilde{O}(\text{nnz}([\widehat{AX}| - \mathbf{y}]) + d^3)$
5: $\{\widehat{\ell}_i([AX| - \mathbf{y}]), \forall i \in [n]\} \leftarrow \frac{(1+\varepsilon)^2}{(1-\varepsilon)^2} \cdot S$
6: **return** $\widehat{\ell}_i([AX| - \mathbf{y}]), \forall i \in [n]$

---

## 5 Empirical Results

We support our theoretical findings via numerical experiments[3] to validate the performance of Algorithm 1 and Algorithm 2 in a non-asymptotic setting.

**Dataset and evaluation methods:** We consider benchmark datasets from the Open Graph Benchmark (OGB) [29], Stanford Network Analysis Project (SNAP) [30], and House dataset [31]. We defer detailed explanations of the datasets to Appendix H. For evaluation methods, we compute the mean squared error (MSE), wall-clock run-time, and peak memory usage of our two proposed schemes and five baselines: (1) REGRESSIONSOLVER$(AX, \mathbf{y})$ with fully known $A$ and $X$; (2) REGRESSIONSOLVER$(\tilde{A}X, \mathbf{y})$ with parts of rows of $A$, say $\tilde{A}$, obtained from sampling rows of $A$ uniformly at random; (3) REGRESSIONSOLVER$(\tilde{A}X, \mathbf{y})$ with the exact leverage score sampling for sampling rows of $A$ to obtain partial $\tilde{A}$; (4) REGRESSIONSOLVER$(\tilde{A}X, \tilde{\mathbf{y}})$ with Algorithm 1; (5) REGRESSIONSOLVER$(\tilde{A}X, \tilde{\mathbf{y}})$ with Algorithm 2; (6) REGRESSIONSOLVER$(\tilde{A}X, \tilde{\mathbf{y}})$ with a sampling scheme inspired by GraphSage [6] that selects the neighborhood of feature aggregation for each node; and (7) REGRESSIONSOLVER$(\tilde{A}X, \tilde{\mathbf{y}})$ with a sampling scheme motivated by GraphSaint [32] that selects a subgraph of the original graph. We note that (6) and (7) are not the tools themselves [6, 32], but graph subsampling strategies inspired by these tools.

**Results and analysis:**
**Mean squared error:** Figure 2 plots the mean squared error with respect to the budget allocation. First, we observe that Algorithm 1 and Algorithm 2 have a similar performance to the one with exact leverage scores sampling, and the performance of the two proposed schemes tends to be similar to that of REGRESSIONSOLVER with full $(AX, \mathbf{y})$, as the budget increases. Particularly seen from Figure 2a, the error reduction on Algorithm 2 from Algorithm 1 is 84% when the budget is at 3%. Even for the sparse datasets, as seen from Figure 2b and Figure 2c, the corresponding improvements of 86% and 67% exist when the budget is at 3% and 7% respectively. We also observe that the error decrease for Algorithm 2 over Algorithm 1; for instance Figure 2b demonstrates the existence of 91% improvement at 5% observation budget. On the other hand, we observe that other subsampling approaches—including those employed in uniform sampling, GraphSage, and GraphSaint—do not perform the graph-weighted regression task as well, especially in the low-budget regime.

**Extension to nonlinear GCN:** While our theoretical results are for linear GCNs, we also assess the efficacy of our proposed algorithm on nonlinear GCNs. Using the ReLU activation function and one hidden layer, Figure 2d plots the mean squared error as a function of the observation budget, shown as a percentage of the number of observed nodes in the graph. Interestingly, we observe that running the non-linear GCN with uniformly sampled $(AX, \mathbf{y})$ fails to perform the graph-weighted regression task, especially in the low-budget regime. On the other hand, we observe that running the non-linear GCN with our proposed sampling technique has a similar performance to the one with exact leverage scores sampling, and the performance of the two proposed schemes tends to be similar to that of Regression with exact $(AX, \mathbf{y})$, as the budget increases. Particularly, the error reduction on Algorithm 1 from Regression with uniformly sampled $(AX, \mathbf{y})$ is 60% when the budget is at 5%.

---

[3]Our code is publicly available online at `https://github.com/seiyun-shin/gnn_node_subsampling`.

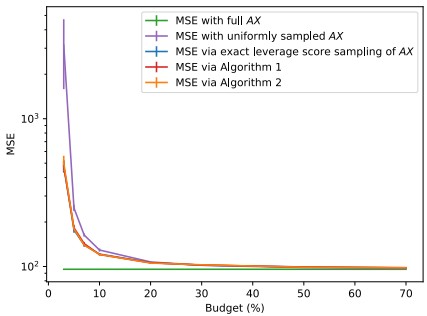

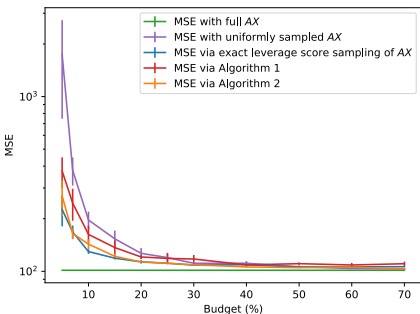

(a) Linear GCN on ogbl-ddi dataset [29] and $X$ generated according to the Cauchy distribution.

(b) Linear GCN on ego-Facebook dataset [30] and $X$ generated by the Cauchy distribution.

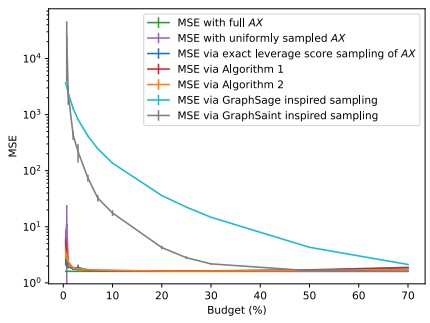

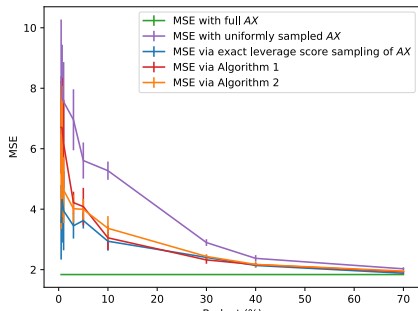

(c) Linear GCN on house dataset [33].

(d) Non-linear GCN on house dataset [33].

Figure 2: MSE w.r.t. observation budget (%): (1) Regression with full $AX$; (2) one with partially observed $AX$ obtained by sampling rows of $A$ uniformly at random; (3) one with partial $AX$ based on the exact leverage score sampling; (4) and (5) one with Algorithm 1 and Algorithm 2; and (6) and (7) ones with GraphSage [6] inspired sampling and GraphSaint [32] inspired sampling algorithms.

Table 1: Wall-clock time comparison on end-to-end processes

| Dataset | # Nodes | # Edges | # Features | Wall-clock time (sec) | |
| --- | --- | --- | --- | --- | --- |
| | | | | use of full $AX$ | use of Algorithm 1 and leverage score sampling |
| ogbl-ddi [29] | 4.3K | 1.3M | 100 | 1.49 | 1.39 |
| ogbn-arxiv [29] | 169.3K | 1.2M | 128 | 299.48 | 7.40 |
| Synthetic data (Gaussian) | 50.0K | 625.0M | 500 | 27.28 | 5.77 |
| Synthetic data (Gaussian) | 100.0K | 2.5B | 500 | 107.10 | 8.97 |
| Synthetic data (Gaussian) | 150.0K | 5.6B | 500 | 247.70 | 9.96 |

**Wall-clock time:** In Table 1, we provide a run-time comparison for the end-to-end training process for a regression task. We compare the wall-clock time of performing a regression task with full $AX$ computation and that with Algorithm 1 that uses partial observations of $A$ and $X$. The results show that our proposed scheme requires orders of magnitude less wall-clock time for large-scale graphs than the regression with exact $AX$ computation. In particular, for the ogbn-arxiv dataset [29], our algorithm runs about 40x faster than the regression with the exact computation of $AX$.

**Peak memory usage:** We also demonstrate the efficacy on the memory usage of our proposed algorithms. See Table 3 in Appendix H for details.

## 6 Concluding Remarks

Motivated by the prohibitive computational/storage costs of running GNNs with large-scale graphs, we considered the problem of subsampling nodes' information for performing a regression task.

Our main contribution lies in providing an efficient sampling algorithm for learning linear GNNs with a large number of nodes with a theoretical guarantee. Specifically, for a two-layer linear GNN with adjacency matrix $A \in \mathbb{R}^{n \times n}$ and data matrix $X \in \mathbb{R}^{n \times d}$, we show that it is possible to learn the model accurately while observing only $O(nd \log n)$ entries of $A$. This in turn yields a run-time of $O(nd^2 \log n)$, avoiding the computation time $O(n^2 d)$ of using the full matrix multiplication $AX$. While we view our main contribution to be of a theoretical flavor, on real-world benchmark datasets, we also demonstrate the run-time/memory improvements of our proposed schemes over other baselines via wall-clock time and peak-memory usage comparisons and MSE comparisons with respect to the observation budget.

## 7    Discussion and Future Work

Our result on learnability with the run-time complexity reduction is particularly significant when the number of nodes $n$ is large and the feature dimension is much smaller compared to $n$ (i.e., $d \ll n$). In addition, we note that the result holds even when the graph adjacency matrix and the data matrix are dense (e.g., weighted graphs where the $a_{ij}$ entries of the adjacency matrix are non-negative real values, and not restricted to $\{0, 1\}$). Examples include pairwise similarity matrices and Erdos-Renyi graphs with constant edge probability $p$, which have a number of edges that scales as $n^2$. In such cases, sparse matrix multiplication would not help much, and our approach could be very useful.

Our proposed scheme is not only useful for speeding up the training of linear GCNs. It is also useful in partial observation settings where the algorithm only has access to a partial view of the graph, and we provide theoretical guarantees that this partial view is still sufficient to accurately perform the regression task. On the contrary, conventional speed-up methods such as sparse matrix multiplication are purely a technique for computational purposes and still require complete information on $A$ and $X$. For instance, consider a social network setting where, in order to know friendships between users, a learner needs to use an API to query the friendships. Note that one needs fewer queries for using our method (since it requires partial $A$) than for using the sparse matrix multiplication (which needs full knowledge of $A$). If the graph is large, the query complexity gain becomes larger.

Several extensions of our results are possible. First, establishing theoretical guarantees for the non-linear GCN framework, and for other target tasks, such as graph link prediction and node classification, would be of significant interest. Based on the empirical observation that our training scheme works well for a two-layer non-linear GCN, extending our theoretical results to the non-linear GCN under some assumptions (as in [34]) on non-linear activation functions seems to be promising. Note that our proposed sampling scheme (based on approximating the leverage score) and the theoretical guarantee we made are tailored for the node-level regression task in the training phase. The reason we focused on this specific setting was so that we could take advantage of the theoretical results on leverage score sampling and linear regression in order to obtain theoretical results for GNN training using a subsampled graph. In particular, we are helped by the fact that in the linear GCN setting, the optimal solution for the linear regression is a matrix-vector product which can be expressed as $((AX)^{\top}(AX))^{\dagger}(AX)^{\top}\mathbf{y}$, where $A$, $X$, and $\mathbf{y}$ denote the adjacency matrix, data matrix, and the labels respectively.

For graph link prediction and classification tasks, on the other hand, since the optimal solution cannot be expressed as a matrix-vector product (due to non-linearities and different cost functions), our proposed algorithm does not extend to the classification problems in a straightforward way, and similarly, our theoretical guarantee may not hold anymore. For future work, we want to study whether our same subsampling process can have any guarantee for a node classification task. For more general tasks, we believe that we need different sampling strategies other than the leverage score sampling and hence we may need other tools for developing a theoretical guarantee for sampling algorithms.

Another direction for future work is to establish generalization guarantees beyond the approximation errors [35]. Lastly, motivated by the fast acceleration of gradient descent algorithms [24–28] and iterative schemes that sample a subset of the entries of $A$ [21, 36], developing algorithms that adaptively select a subset $(A, X)$ to observe based on the information at each gradient descent step would be an interesting direction to explore.

## Acknowledgments

We thank Ruosong Wang, Yun Yang for the helpful discussion and the anonymous reviewers for their constructive feedback. Han Zhao and Seiyun Shin were partly supported by the Defense Advanced Research Projects Agency (DARPA) under Cooperative Agreement Number: HR00112320012, an IBM-IL Discovery Accelerator Institute research award, and Amazon AWS Cloud Credit. The work of Ilan Shomorony was supported in part by the National Science Foundation (NSF) under grant CCF-2046991.

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

# A  Related Work

**Leverage score sampling for linear regression:**   The history of leverage scores traces back to linear regression problems in statistics [37, 38]. The statistical leverages of a matrix indicate the importance of rows in solving a least-squares problem. It has been well-known that leverage score sampling [39, 40] uses leverage scores to sample a subset of the rows of the matrix and solve the regression task with $(1 + \varepsilon)$ accuracy using only $O(d\varepsilon^{-2} \log d)$ subsampled rows of the data matrix $X$ and in time $O(nd^2 + d^3)$, and there have been extensive works on reducing the time complexity as well as the sample complexity [41, 18–22]. However, we note that our setting is a graph-weighted linear regression, which involves a matrix multiplication of $A$ and $X$, taking $O(n^2 d)$ time. To the best of our knowledge, our work is the first one that addresses the setting of graph-weighted linear regression and proposes an algorithm with subquadratic time complexity in $n$.

**Sampling on Graph Convolutional Networks (GCNs):**   Most of the related works focus on reducing the receptive field size via subsampling neighbors of nodes in the input graph.  The motivation is that the neighborhood size can grow exponentially with respect to the number of layers due to the message passing operation of each layer.  In contrast, we focus on the effect of partial observation in one hidden layer and show that node subsampling can reduce the run time as well as the sample complexity while ensuring the desired performance guarantee. We also highlight that our work provides a theoretical guarantee, whereas many of the cited works below provide empirical results without a theoretical guarantee. For readers interested in works related to sampling on GCNs, we refer to the comprehensive survey in [42]. Here we provide a brief summary.

To mitigate the "neighbor explosion" problem, most of the related works [6, 16, 15, 43] take layer-wise sampling approaches to sample neighbors of the nodes in the previous layer in a top-down manner. Specifically, layer-wise samplers [16, 15, 43] adopt importance sampling to sample neighbors in the lower layer given each node in the upper layer, assuming that the sample size for each layer is independent of each other.

Second, node-wise samplers [6] randomly sample neighbors in the lower layer given each node in the upper layer. In addition, [17] proposed a bandit-based approach to perform node-wise sampling in order to optimize the variance in the vein of layer sampling approaches.  See also [44], which proposed a variance-reduced estimator to reduce the receptive field of a single node.

Third, there have been several works [45, 32] that adopt graph-wise sampling approaches.  In particular, the proposed schemes first partition the node set of a graph [45] or sample subgraphs [32], and then train models on those partitions or subgraphs in a batch mode [3]. They show that the training time of each epoch may be much faster compared with layer-wise sampling approaches.

**Graph-weighted linear regression:**   We note that there have been a few works that address linear regression on network-linked data [46–48]. Yet neither sample complexity analysis nor time complexity analysis is provided.

# B  Basic Algorithms

---
**Algorithm 3** LEVERAGESCORE($X$)

---
**Input:** Data matrix $X \in \mathbb{R}^{n \times d}$
**Output:** Leverage score $\ell_i(X)^4$, $\forall i \in [n]$

  1: Compute $K := X^\top X$                                                          $\triangleright O(nd^2)$
  2: $(V, \Sigma, V^\top) \leftarrow \text{SVD}(K)$                                             $\triangleright O(d^3)$
  3: $U \leftarrow X(V\Sigma^{-1})$                                            $\triangleright O(nd^2)$
  4: **return** $\ell_i(X) := \|U_{i:}\|_2^2$, $\forall i \in [n]$                           $\triangleright O(nd)$

---

---

[4]The time complexity is $O(nd^2 + d^3)$ in total, and more precisely, it is $\tilde{O}((\text{nnz}(AX) + d^3) \log(\varepsilon^{-1}))$; see [18–22] for details.

**Algorithm 4** LEVERAGESCORESAMPLING$(A, \{\widehat{\ell}_i([AX| - \mathbf{y}]), \forall i \in [n]\})$

---

**Input:** Adjacency matrix $A \in \mathbb{R}^{n \times n}$, leverage score estimate of $AX$ $\{\widehat{\ell}_i([AX| - \mathbf{y}]), \forall i \in [n]\}$
**Output:** $\tilde{A} \in \mathbb{R}^{O(d\varepsilon^{-2} \log n) \times n}$, scaled labels $\tilde{\mathbf{y}} \in \mathbb{R}^{O(d\varepsilon^{-2} \log n)}$

1: Draw $I_j \sim \text{Bern}(p_j)$, $\forall j \in [n]$ independently, where $p_j = c\widehat{\ell}_j([AX| - \mathbf{y}])\varepsilon^{-2} \log n$
2: $\tilde{A} \leftarrow \sum_{j=1}^{n} \frac{I_j}{\sqrt{p_j}} \mathbf{e}_j A_{j:}$
3: $\tilde{\mathbf{y}} \leftarrow \sum_{j=1}^{n} \frac{I_j}{\sqrt{p_j}} y_j \mathbf{e}_j$
4: **return** $(\tilde{A}, \tilde{\mathbf{y}})$

---

**Algorithm 5** REGRESSIONSOLVER$(\tilde{A}X, \tilde{\mathbf{y}})$

---

**Input:** Matrix $\tilde{A}X \in \mathbb{R}^{O(d\varepsilon^{-2} \log n) \times d}$, scaled labels $\tilde{\mathbf{y}} \in \mathbb{R}^{O(d\varepsilon^{-2} \log n)}$
**Output:** Weight vector $\tilde{\mathbf{w}} \in \mathbb{R}^d$

1: $\tilde{\mathbf{w}} \leftarrow \left( (\tilde{A}X)^\top (\tilde{A}X) \right)^\dagger (\tilde{A}X)^\top \tilde{\mathbf{y}}$
2: **return** $\tilde{\mathbf{w}}$

---

## C  Proof of Proposition 2

Define $t := \max_{\mathbf{v}} \frac{(\mathbf{x}_i^\top \mathbf{v})^2}{\|X\mathbf{v}\|_2^2}$. Hence, it follows that

$$\forall \mathbf{v}, \mathbf{v}^\top \left( \mathbf{x}_i \mathbf{x}_i^\top \right) \mathbf{v} \leq t \cdot \mathbf{v}^\top \left( X^\top X \right) \mathbf{v}.$$

Equivalently, this means that $t$ is the smallest value such that

$$\mathbf{x}_i \mathbf{x}_i^\top \preceq t \cdot X^\top X.$$

Let $\mathbf{w} = X\mathbf{v}$. Since $X$ has full column rank, we know that $\mathbf{v} = X^\dagger \mathbf{w}$. Without loss of generality, assume $\|\mathbf{w}\|_2 = 1$, then we can equivalently transform the above constraint as

$$\forall \mathbf{w}, \|\mathbf{w}\|_2 = 1, \quad \mathbf{w}^\top (X^\dagger)^\top \mathbf{x}_i \mathbf{x}_i^\top X^\dagger \mathbf{w} \leq t.$$

However, note that $(X^\dagger)^\top \mathbf{x}_i \mathbf{x}_i^\top X^\dagger$ is a rank-one matrix, hence it suffices to choose $t$ such that

$$t \geq \text{tr} \left( (X^\dagger)^\top \mathbf{x}_i \mathbf{x}_i^\top X^\dagger \right) = \text{tr} \left( \mathbf{x}_i^\top X^\dagger (X^\dagger)^\top \mathbf{x}_i \right) = \text{tr} \left( \mathbf{x}_i^\top (X^\top X)^\dagger \mathbf{x}_i \right)$$

$$= \text{tr} \left( \mathbf{x}_i^\top (X^\top X)^\dagger \mathbf{x}_i \right) = \mathbf{x}_i^\top (X^\top X)^\dagger \mathbf{x}_i = \ell_i(X).$$

This completes the proof.

## D  Proof of Lemma 1

In this section, we prove Lemma 1, which yields a good approximation by sampling with probability proportional to the leverage score overestimates:

**Lemma 3.** Given a data matrix $X = (\mathbf{x}_1^\top; \ldots; \mathbf{x}_n^\top)^\top \in \mathbb{R}^{n \times d}$ and a vector $\mathbf{u} = (u_1, \ldots, u_n)$ of leverage score *overestimates* of $X$ (i.e., $\ell_i(X) \leq u_i$, $\forall i \in [n]$), suppose we create a matrix $\tilde{X}$ by including the $i$th row of $X$ in $\tilde{X}$ with probability $p_i = \min[1, cu_i\varepsilon^{-2} \log n]$ for some constant $c$, scaled by $1/\sqrt{p_i}$. With probability at least $1 - n^{-\Omega(1)}$, for any $\mathbf{v}$, we have

$$(1 - \varepsilon) \cdot \|X\mathbf{v}\| \leq \|\tilde{X}\mathbf{v}\| \leq (1 + \varepsilon) \cdot \|X\mathbf{v}\|.$$

To prove the lemma, we rely on the following matrix Chernoff bound, which is a variant of the matrix Chernoff bound in [21, Lemma 11] (also, see [49]).

Suppose $W_1, \ldots, W_n$ are independent, random positive semidefinite matrices with size $d \times d$. Define $Z := \sum_{j=1}^{n} W_j$. If $W_j \preceq R \cdot \mathbb{E}[Z]$, then

$$\Pr \left( Z \succeq (1 + \varepsilon)\mathbb{E}[Z] \right) \leq d \cdot \exp \left( -\frac{\varepsilon^2}{3R} \right); \tag{12}$$

$$\Pr \left( Z \preceq (1 - \varepsilon)\mathbb{E}[Z] \right) \leq d \cdot \exp \left( -\frac{\varepsilon^2}{2R} \right), \tag{13}$$

where $A \preceq B$ is equivalent to saying $\mathbf{v}^\top (A - B)\mathbf{v} \geq 0$ for matrices $A$ and $B$ and for all $\mathbf{v}$.

To apply this inequality, define $W_j := \frac{1}{p_j}\mathbf{x}_j\mathbf{x}_j^\top$, which is sampled with probability $p_j$. We first know that $Z$ is an unbiased estimate of $X^\top X$ since

$$\mathbb{E}[Z] = \sum_{j=1}^n p_j \cdot \frac{1}{p_j}\mathbf{x}_j\mathbf{x}_j^\top = \sum_{j=1}^n \mathbf{x}_j\mathbf{x}_j^\top = X^\top X.$$

First, consider the case where $p_j < 1$. This implies that

$$\ell_j(X) \leq u_j \leq \frac{1}{\varepsilon^{-2}\log n}.$$

In order to invoke the matrix Chernoff bound, we only need to verify that $W_j$ $(\succeq 0)$ is spectrally upper bounded:

$$W_j = \frac{1}{cu_j\varepsilon^{-2}\log n}\mathbf{x}_j\mathbf{x}_j^\top \preceq \frac{1}{c\ell_j(X)\varepsilon^{-2}\log n}\mathbf{x}_j\mathbf{x}_j^\top \preceq \frac{\ell_j(X) \cdot X^\top X}{c\ell_j(X)\varepsilon^{-2}\log n} = \frac{X^\top X}{c\varepsilon^{-2}\log n}, \quad (14)$$

where the last inequality holds due to the proof of Proposition 2. Now, the statement follows from the matrix Chernoff bound with $W_j' = (X^\top X)^{-1/2}W_j(X^\top X)^{-1/2}$ and $R = \frac{1}{c\varepsilon^{-2}\log n}$, to conclude that

$$(1 - \varepsilon) \cdot X^\top X \preceq \sum_{j=1}^n W_j \preceq (1 + \varepsilon) \cdot X^\top X,$$

with probability at least $1 - d(n^{-c/3} + n^{-c/2}) = 1 - n^{-\Omega(1)}$, for some $c > 0$ and the regime where $d < n$. Note that we use a union-bound technique to derive the desired high probability guarantee.

Now consider the case where $p_j = 1$, where $W_j := \mathbf{x}_j\mathbf{x}_j^\top$ is sampled with probability 1. In this case, (14) generally does not hold. However, since sampling $W_j$ with probability 1 is equivalent to selecting $W_j^{(1)}, \ldots, W_j^{(c\varepsilon^{-2}\log n)}$, each equal to $\frac{1}{c\varepsilon^{-2}\log n}\mathbf{x}_i\mathbf{x}^\top$ with probability 1, it follows that

$$Y_j^{(i)} \preceq \frac{X^\top X}{c\varepsilon^{-2}\log n}, \quad i = 1, \ldots, c\varepsilon^{-2}\log n.$$

Here we assume that $c\varepsilon^{-2}\log n$ is an integer for simplicity. Replacing $Y_j$ with $\sum_{i=1}^{c\varepsilon^{-2}\log n} Y_j^{(i)}$ yields the desired result since it does not change $\mathbb{E}[Z]$. This completes the proof.

## E  Proof of Theorem 1

We will use the following statement of Matrix Bernstein's inequality [49, Theorem 6.1.1]: Suppose $W_1, \ldots, W_n$ are independent, random matrices with common dimension $d_1 \times d_2$ satisfying

$$\mathbb{E}[W_j] = \mathbf{0}, \quad \text{and} \quad \|W_j\| \leq L, \; \forall j \in [n].$$

Define $Z := \sum_{j=1}^n W_j$. Then for all $t > 0$,

$$\mathbb{E}\left[\|Z\|\right] \leq \sqrt{2v(Z)\log(d_1 + d_2)} + \frac{1}{3}\log(d_1 + d_2); \quad (15)$$

$$\Pr\left(\left\|\sum_{j=1}^n W_j\right\| \geq t\right) \leq (d_1 + d_2) \cdot \exp\left(-\frac{\frac{1}{2}t^2}{v(Z) + \frac{1}{3}Lt}\right), \quad (16)$$

where

$$v(Z) = \max\left\{\left\|\mathbb{E}\left[ZZ^\top\right]\right\|, \left\|\mathbb{E}\left[Z^\top Z\right]\right\|\right\} = \max\left\{\left\|\sum_{j=1}^n \mathbb{E}\left[W_jW_j^\top\right]\right\|, \left\|\sum_{j=1}^n \mathbb{E}\left[W_j^\top W_j\right]\right\|\right\}.$$

To apply this inequality, first recall that Algorithm 1 samples $A_{:j}X_{j:}$ to get a rank-1 matrix $\widehat{A_{:j}X_{j:}} := \frac{I_j}{p_j}A_{:j}X_{j:}$ with probability

$$p_j = \min\left[\frac{B}{n}, 1\right].$$

Defining $W_j := \widehat{A_{:j}X_{j:}} - A_{:j}X_{j:}$, we note that $\mathbb{E}[W_j] = 0$ as shown in (6). In addition, assuming that entries of $A \in \mathbb{R}^{n \times n}$ and $X \in \mathbb{R}^{n \times d}$ satisfy $|A_{ij}| \leq M$ and $|X_{ij}| \leq M$, we then have

$$
\begin{aligned}
\|W_j\| = \left\|\frac{I_j}{p_j}A_{:j}X_{j:} - A_{:j}X_{j:}\right\| &= \left\|\left(\frac{I_j}{p_j} - 1\right)A_{:j}X_{j:}\right\| \\
&\leq \frac{1}{p_j}\|A_{:j}X_{j:}\| \\
&= \max\left[\frac{n}{B}, 1\right] \cdot \|A_{:j}\| \cdot \|X_{j:}\| \\
&\overset{(i)}{=} \frac{n}{B}\|A_{:j}\|\|X_{j:}\| \\
&\overset{(ii)}{\leq} \frac{n^{3/2}d^{1/2}M^2}{B},
\end{aligned}
$$

where $(i)$ follows from the fact that we are interested in the regime where $\frac{B}{n} = o(1)$; and $(ii)$ follows from the fact that $\|A_{:j}\| \leq \sqrt{n}M$ and $\|X_{j:}\| \leq \sqrt{d}M$. We can also bound $\|\mathbb{E}[Z^\top Z]\|$ as

$$
\begin{aligned}
\|\mathbb{E}[Z^\top Z]\| = \left\|\sum_{j=1}^{n}\mathbb{E}\left[W_j^\top W_j\right]\right\| &\leq \sum_{j=1}^{n}\left\|\mathbb{E}\left[W_j^\top W_j\right]\right\| \\
&= \sum_{j=1}^{n}\left\|\mathbb{E}\left[\left(\frac{I_j}{p_j} - 1\right)^2 X_{j:}^\top A_{:j}^\top A_{:j}X_{j:}\right]\right\| \\
&\leq \sum_{j=1}^{n}\frac{1}{p_j}\|X_{j:}\|^2\|A_{:j}\|^2 \\
&\leq \frac{n^3 d M^4}{B},
\end{aligned}
$$

and $\mathbb{E}[ZZ^\top]$ can be similarly bounded, implying that $v(Z) \leq \frac{n^3 d M^4}{B}$. Setting $t = \varepsilon\|AX\|$ yields

$$
\begin{aligned}
\Pr\left(\left\|\sum_{j=1}^{n}W_j\right\| \geq \varepsilon \cdot \|AX\|\right) &\leq 2(n+d)\cdot\exp\left(-\frac{\frac{1}{2}\varepsilon^2\|AX\|^2}{\frac{n^3 d M^4}{B} + \frac{1}{3}\frac{n^{3/2}d^{1/2}M^2}{B}\varepsilon\|AX\|}\right) \\
&= 2(n+d)\cdot\exp\left(-\frac{\frac{1}{2}\varepsilon^2 B}{\frac{n^3 d M^4}{\|AX\|^2} + \frac{1}{3}\frac{n^{3/2}d^{1/2}M^2}{\|AX\|}\varepsilon}\right) \\
&\overset{(iii)}{\leq} 2(n+d)\cdot\exp\left(-\frac{\frac{1}{2}\varepsilon^2 B}{M^4 + \frac{1}{3}M^2\varepsilon}\right) \\
&\overset{(iv)}{\leq} 4n\cdot\exp\left(-\frac{\frac{1}{2}\varepsilon^2 B}{M^4 + \frac{1}{3}M^2\varepsilon}\right),
\end{aligned}
$$

where $(iii)$ follows from Assumption 1 that $\|AX\| \geq n^{3/2}d^{1/2}$; and $(iv)$ follows from the setting that we focus on $d < n$. Hence, as long as we set $B = C\varepsilon^{-2}\log n$ for some constant $C$ ($> 3M^4 > 0$), the exponent of the Matrix Bernstein inequality can be bounded as $-C'\log n$ for some $C' > 0$, and thus can bound the deviation probability to be $n^{-\Omega(1)}$.

Now using the reverse triangle inequality,

$$\left|\|\widehat{AX}\| - \|AX\|\right| \leq \|\widehat{AX} - AX\| \leq \varepsilon\|AX\|,$$

we can conclude that the following spectral approximation holds

$$(1 - \varepsilon)\|AX\| \le \|\widehat{AX}\| \le (1 + \varepsilon)\|AX\|,$$

with probability at least $1 - n^{-\Omega(1)}$. This completes the proof.

# F  Proof of Lemma 2

We will use the following statement of Bernstein's inequality to get the desired result: If $W_1, \ldots, W_n$ are independent zero-mean random variables such that $|W_i| \le M$ with probability 1. Then for all $t > 0$,

$$\Pr\left(\left|\sum_{k=1}^{n} W_k\right| \ge t\right) \le 2\exp\left(-\frac{\frac{1}{2}t^2}{\sum_{i=1}^{n} \mathbb{E}\left[W_i^2\right] + \frac{1}{3}Mt}\right). \tag{17}$$

To apply this inequality, we first provide the following proposition which will serve as key ingredients for proving the lemma.

**Proposition 3.** Fix $(i, j) \in [n] \times [d]$. Let $W_k := \frac{I_k}{p_k} \cdot [A_{:k}X_{k:}]_{ij} - [A_{:k}X_{k:}]_{ij}$. Then for all $k \in [n]$, the following two hold:

1. $|W_k| \le \frac{nM^2}{B}$

2. $\mathbb{E}\left[W_k^2\right] \le \frac{nM^4}{B}$

Note that these two hold for all $(i, j) \in [n] \times [d]$.

*Proof.* We first bound $|W_k|$ as:

$$|W_k| = \left|\frac{I_k}{p_k}[A_{:k}X_{k:}]_{ij} - [A_{:k}X_{k:}]_{ij}\right| = \left|\left(\frac{I_k}{p_k} - 1\right)[A_{:k}X_{k:}]_{ij}\right|$$

$$\le \frac{1}{p_k}\left|[A_{:k}X_{k:}]_{ij}\right|$$

$$= \max\left[\frac{n}{B}, 1\right] \cdot M^2$$

$$\overset{(i)}{=} \frac{nM^2}{B},$$

where $(i)$ follows from the fact that we are interested in the regime where $\frac{B}{n} = o(1)$. We also bound the second moment of $W_k$ as:

$$\mathbb{E}\left[W_k^2\right] = \mathbb{E}\left[\left(\frac{I_k}{p_k} - 1\right)^2 [A_{:k}X_{k:}]_{ij}^2\right]$$

$$\le \frac{1}{p_k}[A_{:k}X_{k:}]_{ij}^2$$

$$\le \frac{nM^4}{B},$$

where we used the facts that $\mathbb{E}[I_{ij}^{(\ell)}] = p_j$. This completes the proof. ∎

Now setting $t = \varepsilon \cdot \mathbf{e}_i^\top AX\mathbf{e}_j$ and applying the Bernstein's inequality now yields:

$$\Pr\left((i, j) \in [n] \times [d] : \left|\sum_{k=1}^{n} W_k\right| \ge t\right)$$

$$= \Pr\left((i, j) \in [n] \times [d] : \left|\mathbf{e}_i^\top \widehat{AX}\mathbf{e}_j - \mathbf{e}_i^\top AX\mathbf{e}_j\right| \ge \varepsilon \cdot \mathbf{e}_i^\top AX\mathbf{e}_j\right)$$

$$\overset{(i)}{\le} n^2 \cdot \Pr\left(\left|\mathbf{e}_i^\top \widehat{AX}\mathbf{e}_j - \mathbf{e}_i^\top AX\mathbf{e}_j\right| \ge \varepsilon \cdot \mathbf{e}_i^\top AX\mathbf{e}_j\right)$$

$$\leq 2n^2 \cdot \exp\left(-\frac{\frac{\varepsilon^2}{2}\left(\mathbf{e}_i^\top AX\mathbf{e}_j\right)^2}{n \cdot \frac{nM^4}{B} + \frac{1}{3} \cdot \frac{nM^2}{B} \cdot \left(\mathbf{e}_i^\top AX\mathbf{e}_j\right)}\right)$$

$$\overset{(ii)}{\leq} 2n^2 \cdot \exp\left(-\frac{\frac{B\varepsilon^2}{2}}{M^4 + \frac{1}{3}M^2}\right),$$

where $(i)$ follows from a union-bound technique; and $(ii)$ follows from Assumption 2 that $\mathbf{e}_i^\top AX\mathbf{e}_j = \Omega(n)$, $\forall (i,j) \in [n] \times [d]$. Setting $B = \frac{C}{\varepsilon^2}\log n$ for sufficiently large $C > \frac{16}{3}M^4 > 0$, one can observe that the last bound is upper bounded by $n^{-\Omega(1)}$. Since the other direction holds similarly, we conclude that with probability $1 - n^{-\Omega(1)}$, with budget of $B = \frac{C}{\varepsilon^2}\log n$,

$$(1 - \varepsilon) \cdot \mathbf{e}_i^\top AX\mathbf{e}_j \leq \mathbf{e}_i^\top \widehat{AX}\mathbf{e}_j \leq (1 + \varepsilon) \cdot \mathbf{e}_i^\top AX\mathbf{e}_j, \quad \forall (i,j) \in [n] \times [d].$$

For the augmented matrix $[AX| - \mathbf{y}]$, the above then yields

$$(1 - \varepsilon) \cdot \mathbf{e}_i^\top [AX| - \mathbf{y}]\mathbf{e}_j \leq \mathbf{e}_i^\top [\widehat{AX}| - \mathbf{y}]\mathbf{e}_j \leq (1 + \varepsilon) \cdot \mathbf{e}_i^\top [AX| - \mathbf{y}]\mathbf{e}_j, \quad \forall (i,j) \in [n] \times [d].$$

Pick $\mathbf{v} \in \mathbb{R}^{d+1}$, where it can be represented as

$$\mathbf{v} = \sum_{k=1}^{d+1} \alpha_k \mathbf{e}_k,$$

for some $\alpha_1, \ldots, \alpha_{d+1}$. From the above, one can readily see that with probability at least $1 - n^{-\Omega(1)}$,

$$\frac{(\mathbf{e}_i^\top [AX| - \mathbf{y}]\mathbf{v})^2}{\|[AX| - \mathbf{y}]\mathbf{v}\|_2^2} \leq \left(\frac{1+\varepsilon}{1-\varepsilon}\right)^2 \cdot \frac{(\mathbf{e}_i^\top [\widehat{AX}| - \mathbf{y}]\mathbf{v})^2}{\|[\widehat{AX}| - \mathbf{y}]\mathbf{v}\|_2^2} \leq \left(\frac{1+\varepsilon}{1-\varepsilon}\right)^4 \cdot \frac{(\mathbf{e}_i^\top [AX| - \mathbf{y}]\mathbf{v})^2}{\|[AX| - \mathbf{y}]\mathbf{v}\|_2^2}, \quad \forall \mathbf{v}.$$

Choosing $\mathbf{v}^* := \arg\max_{\mathbf{v}} \frac{(\mathbf{e}_i^\top [\widehat{AX}| - \mathbf{y}]\mathbf{v})^2}{\|[\widehat{AX}| - \mathbf{y}]\mathbf{v}\|_2^2}$ and $\mathbf{w}^* := \arg\max_{\mathbf{w}} \frac{(\mathbf{e}_i^\top [AX| - \mathbf{y}]\mathbf{w})^2}{\|[AX| - \mathbf{y}]\mathbf{w}\|_2^2}$, from the above, we have

$$\left(\frac{1+\varepsilon}{1-\varepsilon}\right)^2 \cdot \frac{(\mathbf{e}_i^\top [\widehat{AX}| - \mathbf{y}]\mathbf{v}^*)^2}{\|[\widehat{AX}| - \mathbf{y}]\mathbf{v}^*\|_2^2} \leq \left(\frac{1+\varepsilon}{1-\varepsilon}\right)^4 \cdot \frac{(\mathbf{e}_i^\top [AX| - \mathbf{y}]\mathbf{v}^*)^2}{\|[AX| - \mathbf{y}]\mathbf{v}^*\|_2^2} \leq \left(\frac{1+\varepsilon}{1-\varepsilon}\right)^4 \cdot \frac{(\mathbf{e}_i^\top [AX| - \mathbf{y}]\mathbf{w}^*)^2}{\|[AX| - \mathbf{y}]\mathbf{w}^*\|_2^2},$$

where the last formula is $(\frac{1+\varepsilon}{1-\varepsilon})^4 \ell_i(AX)$ by the choice of $\mathbf{w}^*$. In a similar manner, we obtain

$$\frac{(\mathbf{e}_i^\top [AX| - \mathbf{y}]\mathbf{w}^*)^2}{\|[AX| - \mathbf{y}]\mathbf{w}^*\|_2^2} \leq \left(\frac{1+\varepsilon}{1-\varepsilon}\right)^2 \cdot \frac{(\mathbf{e}_i^\top [\widehat{AX}| - \mathbf{y}]\mathbf{w}^*)^2}{\|[\widehat{AX}| - \mathbf{y}]\mathbf{w}^*\|_2^2} \leq \left(\frac{1+\varepsilon}{1-\varepsilon}\right)^2 \cdot \frac{(\mathbf{e}_i^\top [\widehat{AX}| - \mathbf{y}]\mathbf{v}^*)^2}{\|[\widehat{AX}| - \mathbf{y}]\mathbf{v}^*\|_2^2}.$$

Referring to Proposition 2 and the leverage score estimate $\widehat{\ell}_i(AX)$ satisfying

$$\widehat{\ell}_i([AX| - \mathbf{y}]) = \left(\frac{1+\varepsilon}{1-\varepsilon}\right)^2 \ell_i([\widehat{AX}| - \mathbf{y}]) = \left(\frac{1+\varepsilon}{1-\varepsilon}\right)^2 \cdot \max_{\mathbf{v}} \frac{(\mathbf{e}_i^\top [\widehat{AX}| - \mathbf{y}]\mathbf{v})^2}{\|[\widehat{AX}| - \mathbf{y}]\mathbf{v}\|_2^2},$$

we get the desired result of

$$\ell_i([AX| - \mathbf{y}]) \leq \widehat{\ell}_i([AX| - \mathbf{y}]) \leq \left(\frac{1+\varepsilon}{1-\varepsilon}\right)^4 \ell_i([AX| - \mathbf{y}]), \quad \forall i \in [n],$$

This completes the proof.

# G   Theoretical Guarantees for Algorithm 2

In this section, we provide theoretical guarantees for Algorithm 2. As mentioned in Section 4, we obtain the same series of theoretical guarantees as those for Algorithm 1. For completeness, we leave the detailed statements below.

**Theorem 3.** Under Assumption 1, if we run Algorithm 2 with the budget $B = C\varepsilon^{-2}\log n$ for large enough $C > 0$, the estimate $\widehat{AX}$ satisfies

$$(1 - \varepsilon) \cdot \|AX\mathbf{v}\| \leq \|\widehat{AX}\mathbf{v}\| \leq (1 + \varepsilon) \cdot \|AX\mathbf{v}\|, \quad \forall \mathbf{v} \in \mathbb{R}^d \tag{18}$$

with probability at least $1 - n^{-\Omega(1)}$.

**Lemma 4.** Under Assumption 1, Algorithm 2 with the budget $B = C\varepsilon^{-2}\log n$ for large enough $C > 0$ outputs the leverage score overestimates with a $\left(\frac{1+\varepsilon}{1-\varepsilon}\right)^4$ multiplicative ratio; i.e.,

$$\ell_i([AX| - \mathbf{y}]) \leq \widehat{\ell}_i([AX| - \mathbf{y}]) \leq \left(\frac{1+\varepsilon}{1-\varepsilon}\right)^4 \ell_i([AX| - \mathbf{y}]), \quad \forall i \in [n], \tag{19}$$

with probability at least $1 - n^{-\Omega(1)}$.

**Theorem 4.** Under Assumption 1, suppose we run Algorithm 2 with the budget $B = C\varepsilon^{-2}\log n$ for large enough $C > 0$, to obtain $\{\widehat{\ell}_i([AX| - \mathbf{y}]), \forall i \in [n]\}$. Subsequently, if we run LEVERAGESCORESAMPLING$(A, \{\widehat{\ell}_i([AX| - \mathbf{y}]), \forall i \in [n]\})$, we obtain $\tilde{A} \in \mathbb{R}^{O(d\varepsilon^{-2}\log n)\times n}$ and $\tilde{\mathbf{y}} \in \mathbb{R}^{O(d\varepsilon^{-2}\log n)}$ such that REGRESSIONSOLVER$(\tilde{A}X, \tilde{\mathbf{y}})$ provides

$$\min_{\tilde{\mathbf{w}}} \|\mathbf{y} - \tilde{A}X\tilde{\mathbf{w}}\|_2^2 \leq (1+\varepsilon) \cdot \min_{\mathbf{w}} \|\mathbf{y} - AX\mathbf{w}\|_2^2.$$

We note that it suffices to provide the proof of Theorem 3, as the rest follows in a similar manner to those for Algorithm 1.

**Proof of Theorem 3:** As in Appendix E, we will use the Matrix Bernstein's inequality described in (16). Recall that Algorithm 2 samples $A_{:j}X_{j:}$ to get a rank-1 matrix $\widehat{A_{:j}X_{j:}} := \frac{I_j}{p_j}A_{:j}X_{j:}$ with probability as in (11)

$$p_j = \min\left[B \cdot \frac{\|A_{:j}\|\|X_{j:}\|}{\sum_j \|A_{:j}\|\|X_{j:}\|}, 1\right],$$

such that $\sum_j p_j \leq B$.

Defining $W_j := \widehat{A_{:j}X_{j:}} - A_{:j}X_{j:}$ and we note that $\mathbb{E}[W_j] = 0$ as shown in (6). In addition, assuming that entries of $A \in \mathbb{R}^{n\times n}$ and $X \in \mathbb{R}^{n\times d}$ satisfy $|A_{ij}| \leq M$ and $|X_{ij}| \leq M$, we then have

$$\|W_j\| = \left\|\frac{I_j}{p_j}A_{:j}X_{j:} - A_{:j}X_{j:}\right\| = \left\|\left(\frac{I_j}{p_j} - 1\right)A_{:j}X_{j:}\right\|$$
$$\leq \frac{1}{p_j}\|A_{:j}X_{j:}\|$$
$$= \max\left[\frac{1}{B}\cdot\frac{\sum_j\|A_{:j}\|\|X_{j:}\|}{\|A_{:j}\|\|X_{j:}\|}, 1\right]\cdot\|A_{:j}\|\cdot\|X_{j:}\|$$
$$= \max\left[\frac{1}{B}\cdot\sum_{j=1}^n\|A_{:j}\|\|X_{j:}\|, \|A_{:j}\|\|X_{j:}\|\right]$$
$$\overset{(i)}{\leq} \max\left[\frac{n^{3/2}d^{1/2}M^2}{B}, n^{1/2}d^{1/2}M\right]$$
$$\overset{(ii)}{\leq} \frac{n^{3/2}d^{1/2}M^2}{B},$$

where $(i)$ follows from the fact that $\|A_{:j}\| \leq \sqrt{n}M$ and $\|X_{j:}\| \leq \sqrt{d}M$; and $(ii)$ follows from the fact that we are interested in the regime where $B = o(n)$. We can also bound $\|\mathbb{E}[Z^\top Z]\|$ as

$$\|\mathbb{E}[Z^\top Z]\| = \left\|\mathbb{E}\left[\sum_{j=1}^n\left(\frac{I_j}{p_j}A_{:j}X_{j:} - A_{:j}X_{j:}\right)^\top\left(\frac{I_j}{p_j}A_{:j}X_{j:} - A_{:j}X_{j:}\right)\right]\right\|$$
$$= \left\|\sum_{j=1}^n\left[\frac{1}{p_j^2}(A_{:j}X_{j:})^\top(A_{:j}X_{j:})\cdot p_j - 2(A_{:j}X_{j:})^\top(A_{:j}X_{j:}) + (A_{:j}X_{j:})^\top(A_{:j}X_{j:})\right]\right\|$$

$$= \left\| \sum_{j=1}^{n} \left( \frac{1}{p_j} - 1 \right) \cdot \left( \|A_{:j}\|^2 \cdot X_{j:}^{\top} X_{j:} \right) \right\|$$

$$\leq \left\| \sum_{j=1}^{n} \left[ \frac{1}{B} \cdot \left( \frac{\sum_{j=1}^{n} \|A_{:j}\| \|X_{j:}\|}{\|A_{:j}\| \|X_{j:}\|} \right) \cdot \left( \|A_{:j}\|^2 \cdot X_{j:}^{\top} X_{j:} \right) \right] \right\|$$

$$\leq \frac{1}{B} \cdot \left( \sum_{j=1}^{n} \|A_{:j}\| \|X_{j:}\| \right) \cdot \left\| \sum_{j=1}^{n} \frac{\|A_{:j}\|}{\|X_{j:}\|} X_{j:}^{\top} X_{j:} \right\|$$

$$= \frac{1}{B} \cdot \left( \sum_{j=1}^{n} \|A_{:j}\| \|X_{j:}\| \right)^2$$

$$\leq \frac{n^3 d M^4}{B}.$$

Also, one can readily see that $\mathbb{E}[ZZ^{\top}]$ can be similarly bounded, implying that $v(Z) \leq \frac{n^3 d M^4}{B}$. Setting $t = \varepsilon \|AX\|$ yields:

$$\Pr\left( \left\| \sum_{j=1}^{n} W_j \right\| \geq \varepsilon \cdot \|AX\| \right) \leq 2(n+d) \cdot \exp\left( -\frac{\frac{1}{2}\varepsilon^2 \|AX\|^2}{\frac{n^3 d M^4}{B} + \frac{1}{3} \frac{n^{3/2} d^{1/2} M^2}{B} \varepsilon \|AX\|} \right)$$

$$\leq 2(n+d) \cdot \exp\left( -\frac{\frac{1}{2}\varepsilon^2 B}{\frac{n^3 d M^4}{\|AX\|^2} + \frac{1}{3} \frac{n^{3/2} d^{1/2} M^2}{\|AX\|} \varepsilon} \right)$$

$$\overset{(iii)}{\leq} 2(n+d) \cdot \exp\left( -\frac{\frac{1}{2}\varepsilon^2 B}{M^4 + \frac{1}{3} M^2 \varepsilon} \right)$$

$$\overset{(iv)}{\leq} 4n \cdot \exp\left( -\frac{\frac{1}{2}\varepsilon^2 B}{M^4 + \frac{1}{3} M^2 \varepsilon} \right),$$

where $(iii)$ follows from Assumption 1 that $\|AX\| \geq n^{3/2} d^{1/2}$; and $(iv)$ follows from the setting that we focus on $d < n$. Hence, as long as we set $B = C\varepsilon^{-2} \log n$ for some constant $C \ (> 3M^4 > 0)$, the exponent of the Matrix Bernstein inequality can be bounded as $-C' \log n$ for some $C' > 0$, and thus can bound the deviation probability to be $n^{-\Omega(1)}$.

Now using the reverse triangle inequality,

$$\left| \|\widehat{AX}\| - \|AX\| \right| \leq \|\widehat{AX} - AX\| \leq \varepsilon \|AX\|,$$

we can conclude that the following spectral approximation holds

$$(1 - \varepsilon)\|AX\| \leq \|\widehat{AX}\| \leq (1 + \varepsilon)\|AX\|,$$

with probability at least $1 - n^{-\Omega(1)}$. This completes the proof.

## H   Further Implementation Details

In this section, we provide additional details on the explanations of the datasets, wall-clock time with the adoption of sparse matrix multiplication, and peak memory usage comparisons.

**Datasets:**   We consider benchmark datasets from the Open Graph Benchmark (OGB) [29], Stanford Network Analysis Project (SNAP) [30], and House dataset [31]. For the MSE comparisons, we consider (1) ogbl-ddi dataset from OGB, (2) ego-Facebook dataset from SNAP, and (3) House dataset. We note that while the third dataset has its data matrix (i.e., a concatenation of feature vectors) and labels, the first two datasets do not have feature matrices and target variables. Accordingly, to conduct controlled experiments, we synthetically generate data matrix $X$, weight vector $\mathbf{w}$, and noisy labels

Table 2: Wall-clock time comparison on end-to-end processes with sparse matrix multiplication

| Dataset | # Nodes | # Edges | # Features | Wall-clock time (sec) | |
|---|---|---|---|---|---|
| | | | | use of full $AX$ | use of Algorithm 1 and leverage score sampling |
| ogbl-ddi [29] | 4.3K | 1.3M | 100 | 1.02 | 0.84 |
| ogbn-arxiv [29] | 169.3K | 1.2M | 128 | 6.69 | 0.86 |
| Synthetic data (Gaussian) | 50.0K | 625.0M | 500 | 2452.14 | 58.30 |

Table 3: Peak memory usage comparison on sampling and regression steps

| Dataset | # Nodes | # Edges | # Features | Peak memory (MiB) | |
|---|---|---|---|---|---|
| | | | | use of full $AX$ | use of Algorithm 1 and leverage score sampling |
| ogbl-ddi [29] | 4.3K | 1.3M | 100 | 1.45 | 0.17 |
| ogbn-arxiv [29] | 169.3K | 1.2M | 128 | 330.75 | 160.45 |
| Synthetic data (Gaussian) | 50.0K | 625.0M | 500 | 240.43 | 0.17 |
| Synthetic data (Gaussian) | 100.0K | 2.5B | 500 | 431.82 | 0.52 |
| Synthetic data (Gaussian) | 150.0K | 5.6B | 500 | 865.25 | 19.39 |

$\mathbf{y}$ by setting the linear relationship between labels and features: $\mathbf{y} := X\mathbf{w} + \mathbf{n}$. Here $\mathbf{n}$ denotes an additive Gaussian noise with parameters $(\mu, \sigma) = (1, 10)$. Based upon the size of the two datasets described below, we consider the data matrices $X \in \mathbb{R}^{n \times d}$ (with $n \in \{4267, 4039\}$ and $d = 100$), where each row (i.e., each node's feature vector) is drawn according to the Cauchy distribution with parameters $(x_0, \gamma) = (10, 100)$, in an i.i.d. manner. Here $x_0$ denotes the location parameter, indicating the location of the peak of the distribution, and $\gamma$ denotes the scale parameter, specifying the half-width at half-maximum (HWHM). Also, we consider $\mathbf{w} \in \mathbb{R}^d$ generated by the Cauchy distribution with $(x_0, \gamma) = (10, 100)$.

**ogbl-ddi dataset [29]:** This dataset represents a undirected and unweighted graph which models the drug-drug interactions. Each node corresponds to an FDA-approved or experimental drug. An edge indicates interactions between drugs that can be interpreted as the difference between the joint effect of taking the two drugs together and the expected effect when drugs act independently of each other. As there are neither feature vectors nor labels that are available, we synthetically generate those as mentioned above.

**ego-Facebook dataset [30]:** This dataset models a social network that represents friendship of "circles" between users from Facebook. Examples of such relationships between users include students of common universities, sports teams, relatives, etc. Notice that the graph adjacency matrix is more sparse compared to ogbl-ddi dataset. As above, since there exists no feature vectors and labels that are available, we synthetically generate those to perform regression task.

**House dataset [31]:** This dataset represents a graph in which nodes correspond to the properties; the existence of edges means that two properties are located close to each other; and the target label is the property's price. In addition, there exist node features representing MedInc, HouseAge, AveRooms, AveBedrms, Population, and AveOccup. In particular, we use the refined dataset from [33].

Furthermore, for the wall-clock time and peak memory usage comparisons, we additionally use ogbl-arxiv [29] dataset, representing the citation network between all Computer Science (CS) arXiv papers (although the dataset is tailored for classification of 40 subject areas) and synthetic dataset where all $(A, X, \mathbf{y})$ are generated according to Gaussian distribution with size specified in Table 1, Table 2, and Table 3.

**Additional results and analysis:**
**Wall-clock time with the adoption of sparse matrix multiplication:** Notice that as compared to results in Table 1, Table 2 demonstrates that the computation time of both our approach and using exact $AX$ become faster due to the adoption of sparse matrix multiplication and the reduction time was larger for using exact $AX$ (especially for the ogbn-arxiv dataset which is sparse). However, we observe that our proposed scheme is faster than the regression by using the exact multiplication of $AX$. Furthermore, for the synthetic dataset where $A$ and $X$ are generated according to a Gaussian

distribution and hence $A$ is dense, we observe that the sparse matrix multiplication does not help speed up the computation time and the matrix multiplication with scipy.sparse package took even longer than the matrix multiplication with numpy package.

**Peak memory usage:** In addition to run-time, we expect our proposed approach to offer significant improvements in terms of memory usage. As a simple experiment in terms of memory usage, we provide the peak memory usage comparison for the same datasets during the sampling and regression steps of the algorithm. As shown in Table 3, in the best case, the proposed algorithm requires 1414× less memory than the algorithm using the exact computation of $AX$.

**Computing architecture for wall-clock time and peak-memory usage comparisons:**  For a fair comparison on Table 1, Table 2, and Table 3, we use the same regression solver and use the same specification of 48 cores of an x86 64 processor with 503.74GB memory.

