# OpenReview forum: "Efficient Learning of Linear Graph Neural Networks via Node Subsampling"
_NeurIPS.cc/2023/Conference — NeurIPS 2023 poster_

### Official Review · Reviewer_XHGm · 2023-06-22

**Soundness:** 3 good
**Presentation:** 2 fair
**Contribution:** 2 fair
**Rating:** 5
**Confidence:** 3

**Summary:**

The work tackles the problem of scaling GNNs used for regression tasks to large graphs by subsampling given nodes. The technique consists of first performing node sub-sampling to estimate the leverage scores of $AX$ and then performs leverage score sampling on $AX$. The authors show that this technique is a good spectral approximation to the full $AX$ computation, but avoids the $O(n^2d)$ matrix multiplication cost.

**Strengths:**

The paper tackles the interesting and practically useful problem of scaling GNNs to large graphs. The proposed technique has useful theoretical guarantees and as far as I am aware it is a novel approach. The approximation to the regression problem is interesting as it has useful spectral guarantees and could be of general interest from an algorithmic perspective outside of the GNN community.

**Weaknesses:**

There are a few weaknesses of the work, some of which the authors point out. The most glaring one is the fact that this work targets only linear GNNs. While analysing non-linear models is clearly more challenging, it would be interesting to have some comments or results for non-linear models with more easy to study non-linearities (such as ReLU).

I found following the different run-times a bit challenging. It may be useful to have a summary table that compares the various computational/memory complexities for the different techniques used in the experiments in Section 5.

As this problem is a practical one, I believe Section 5 could be stronger. While the authors argue that the matrix multiplication is a bottleneck, modern GPUs are extremely optimized for exactly these sort of operations and as a consequence it may be the case that the full matrix multiplication is still faster than sampling. As such, I would be very interested to see some results on the runtimes of the various techniques. Further, it would be interesting to test regimes in which the graph is too large to fit in a GPU and sampling is absolutely necessary.

It may also be useful to run experiments with non-linear GNNs using the proposed sampling technique.

I also spotted some typos and formatting issues in the manuscript (line 60, line 95).

**Questions:**

While the work mentions linear GNNs, it seems to be the case that the work only considers one-hop linear GNNs. While a multi-layer linear GNN is still a linear one, do these results apply to multi-hop linear GNNs as well? It would be useful to have more details on the number of layers and exact model used in section 5 as well.

Is there a reason in figure 2 (b) and (d) that MSE with uniformly sampled $AX$ is not included in the experiments?



**Limitations:**

The authors acknowledge a main limitation that is the fact that this work only studies linear GNNs. A further limitation is that while the authors acknowledge the improved theoretical run-times of their technique, they do not give practical wall-clock runtimes for the experiments. While the technique may be theoretically faster, in practice many operations such as matrix operation are extremely optimized.

---

> ### Author Rebuttal · Authors · 2023-08-10
>
> W1 & W2. Lack of run-time comparison:
>
> Thank you for pointing out this issue. Please see the common response and the additional experiments in the attached pdf above.
>
> W3. Extension to nonlinear GCN:
>
> Please see the common response.
>
> W4. Some typos (line 60, line 95):
>
> Thanks for pointing out these typos. We fixed those.
>
> Q1. Extension to the more-than-two-layer case:
>
> We thank the reviewer for pointing out that we did not discuss this properly. Note that without non-linearities, considering a multi-layer network is equivalent to performing additional matrix multiplication by $A$ and the output of the previous layer. Since our proposed scheme yields a spectral approximation guarantee for matrix multiplication, we believe that an analysis similar to that of Algorithm 1 should be possible for this extension to obtain a $(1+\epsilon)^{L}$-approximation guarantee, where $L$ denotes the number of layers. Since the approximation guarantee comes with high probability, a union bound could be used to obtain a high probability guarantee for the multi-layer case (since the order or error for each approximate matrix multiplication is the same). If the paper is accepted, we will include a discussion on this extension.
>
> Q2. Is there a reason in Figures 2 (b) and (d) that MSE with uniformly sampled AX is not included in the experiments?
>
> Thank you for pointing out the need for clarification. Figures 2b and 2d are the magnified views of 2a and 2c and hence the uniformly sampled curve does not appear within the axes limits, respectively.
>
> L1. The authors acknowledge a main limitation which is the fact that this work only studies linear GNNs. A further limitation is that while the authors acknowledge the improved theoretical run-times of their technique, they do not give practical wall-clock runtimes for the experiments. While the technique may be theoretically faster, in practice many operations such as matrix operation are extremely optimized.
>
> We agree that the focus on linear GNNs is a limitation of this work, but we view our work as studying a specific regime where existing theoretical tools allow the derivation of theoretical guarantees. We also agree that practical wall-clock run-times for the experiments are important here. We have run additional experiments to provide wall-clock runtimes for the experiments and also empirical results on nonlinear GNNs. Please see the common response above and the attached pdf file for detailed experiment results.

---

> > ### Comment · Reviewer_XHGm · 2023-08-14
> >
> > Thank you for the additional information and further experiments. I think the work is an interesting direction and I feel like the experiments with the wall-clock/memory time are indeed promising. The extension to multi-hop GNNs would definitely add to the analysis. I have increased my score accordingly.

---

> > > ### Author Response · Authors · 2023-08-15
> > >
> > > We thank the reviewer for their helpful discussion, feedback, and updated score. Please let us know if you have further questions.

---

### Official Review · Reviewer_F1DR · 2023-06-27

**Soundness:** 4 excellent
**Presentation:** 4 excellent
**Contribution:** 3 good
**Rating:** 8
**Confidence:** 3

**Summary:**

This paper proposes a method to overcome the computational difficulties often encountered when using graph neural networks (GNNs) for large-scale datasets, by subsampling in the adjacency and feature matrices. By assuming a two-layer linear GNN for the regression problem and performing subsampling based on the leverage score, the authors show that the MSE of the reduced problem generated by their proposed method is bounded from above by the true MSE times $(1+\epsilon)$.


**Strengths:**

Originality & Significance: The proposed method basically follows the common idea in subsampling algorithms for linear regression in a standard setting; the same is true for the derivation of the upper bound of the reduced problem MSE. However, in the current problem, the regression coefficients are multiplied with the product of the adjacency matrix and the feature matrix, and the computation of the product itself is expensive and should be avoided. A new twist is introduced at this point: the proposed algorithm breaks the problem into two steps; in the first step the leverage score is approximately estimated by subsampling the rows of the adjacency matrix and the corresponding columns of the feature matrix; in the second step, the reduced adjacency matrix is generated according to the estimated leverage score. In this way, the direct computation of the product is avoided. I think this idea is interesting. Subsampling algorithms with theoretical guarantees are currently rare in this field, and this makes the proposed method valuable. The originality and significance of the proposed method are considered to be thus high.


Quality: The explanation of the algorithm and the derivation of the theoretical guarantee are very clear. The numerical experiment is also convincing. The quality of the paper is thus high.


Clarity: The presentation is nice and clear, except that the references are missing in the main manuscript (they are in the supplementary material). I think the explanations of the numerical experiment result should be more detailed, though the important parts are understandable. Some typos are also found. Although there are small flaws like these, the overall clarity is high.

List of typos found:

Line 95: discussio -> discussion

Line 186: unnecessary space seems to exist before ``Algorithm''.

Line 225: unnecessary space seems to exist before ``Algorithm''.

Line 226: Appendix Appendix G -> Appendix G

Line 245: 4039 -> 4039 nodes

Line 248: 100),in -> 100), in


**Weaknesses:**

- The linear GNN is not employed in practice, and the extension to nonlinear GNN is desired. I think it is at least possible to apply the proposed algorithm to the nonlinear case, although there is no theoretical guarantee in that case. Some more discussion or experimental results in such a nonlinear case are welcome.

- The extension to the more-than-two layer case is also nontrivial. Some more discussions are welcome.


**Questions:**

- Concerning to the experimental section, how about the computational time in practice? The proposed method comes from the computational issue and hence it is better to show the computational time/cost as the result.

- Figures 2b and 2d are the magnified views of 2a and 2c, respectively. Is this right? If so, it is better to mention this point.


**Limitations:**

The authors address their research limitations well. I think there is no concern about the potential societal impact.

---

> ### Author Rebuttal · Authors · 2023-08-10
>
> W1. Extension to nonlinear GCN:
>
> We thank the reviewer for suggesting interesting directions for future work. We leave comments in the common response section and provide additional experimental results for nonlinear GCN in the attached pdf.
>
> W2. Extension to the more-than-two-layer case:
>
> We thank the reviewer for pointing out that we did not discuss this properly. Note that without non-linearities, considering a multi-layer network is equivalent to performing additional matrix multiplication by $A$ and the output of the previous layer. Since our proposed scheme yields a spectral approximation guarantee for matrix multiplication, we believe that an analysis similar to that of Algorithm 1 should be possible for this extension to obtain a $(1+\epsilon)^{L}$-approximation guarantee, where $L$ denotes the number of layers. Since the approximation guarantee comes with high probability, a union bound could be used to obtain a high probability guarantee for the multi-layer case (since the order or error for each approximate matrix multiplication is the same). If the paper is accepted, we will include a discussion on this extension.
>
> Q1. Concerning to the experimental section, how about the computational time in practice? The proposed method comes from the computational issue and hence it is better to show the computational time/cost as the result.
>
> Please see the common response and additional run-time experiments described in the attached pdf.
>
> Q2. Figures 2b and 2d are the magnified views of 2a and 2c, respectively. Is this right? If so, it is better to mention this point.
>
> Yes, that is correct. We will add a sentence to mention that point.

---

> > ### Comment · Reviewer_F1DR · 2023-08-14
> >
> > I thank the authors' responses and additional experiments. These completely address my issues and I am satisfied. I also have read the discussions between the other reviewers and the authors. Although there seem to be some objections against the simple problem setting by the authors, which is different from the practical settings where GNNs are applied, I think, in my opinion, the authors are providing reasonable responses against the criticisms from the reviewers, from the theoretical viewpoint. Thanks to these efforts, I think the paper increases its value. Accordingly, I would like to raise the score.

---

> > > ### Author Response · Authors · 2023-08-15
> > >
> > > We sincerely appreciate your acknowledgment of our paper and the updated score. Based on the discussion and feedback, we will keep polishing the revision to a better version.

---

### Official Review · Reviewer_J15Y · 2023-06-28

**Soundness:** 3 good
**Presentation:** 2 fair
**Contribution:** 2 fair
**Rating:** 4
**Confidence:** 4

**Summary:**

The authors proposed a sampling method to train graph neural networks efficiently. However, the current implementation of graph neural networks is based on the sparse matrix multiplication and thus the complexity is not O(n^2d). The authors only analyze the complexity theoretically without experimental supports. Both of major issues make the contribution weak.

**Strengths:**

1. Theoratically, the proposed method is efficient compared with the dense implementation of graph neural networks.
2. The experiments are conducted on the large scale datasets with different variant of the methods.

**Weaknesses:**

1. The statement about the complexity of the graph neural network is not correct. The current implementation is based on the sparse multiplication, so the complexity is not O(n^2d).
2. The computational time comparision is missing, which is essential for this work.
3. There is lack of baselines, it only compares with its own variants.


**Questions:**

1. What is the computational time of this approach compared with sparse matrix multiplication?
2. Besides MSE, how about applying this to the real semi-supervised classification tasks? Will this lead to the same prediction accuracy?
3. Is it possible to use this for link prediction tasks when dropping edges for validation? Is there still any theory guarantee in this case?

---

> ### Author Rebuttal · Authors · 2023-08-10
>
> W1. The statement about the complexity of the graph neural network is not correct. The current implementation is based on sparse multiplication, so the complexity is not O(n^2d).
>
> We thank the reviewer for pointing this out. The main motivation for our graph subsampling procedure is settings where the adjacency matrix is dense (with $\Theta(n^2)$ non-zero entries), and sparse multiplication would still require time $O(n^2d)$. The reviewer is correct that, whenever $A$ is sparse, sparse matrix multiplication could avoid the $O(n^2d)$ complexity. We will correct the paper to be clear on this point.
>
> W2. Lack of run-time comparison:
>
> We thank the reviewer for pointing out this issue. Please see the common response and the attached pdf file for further experiments.
>
> W3. Comparison with other baselines:
>
> Thanks for the suggestion. Please see the common response and the attached pdf file for details.
>
> Q1. What is the computational time of this approach compared with sparse matrix multiplication?
>
> We believe that a fair comparison would not be comparing our proposed algorithm with sparse matrix multiplication since our algorithm includes a multiplication step after reducing the size of the graph adjacency matrix via leverage score sampling. Hence our algorithm could potentially be combined with sparse matrix multiplication. For this reason, we would expect a similar tendency as demonstrated in Table 1 in the attached pdf that even with adopting sparse matrix multiplication; that is, the computation time of our approach will be faster than the regression by using the exact multiplication of $AX$.
>
> Q2. Besides MSE, how about applying this to the real semi-supervised classification tasks? Will this lead to the same prediction accuracy?
>
> We thank the reviewer for suggesting interesting directions for future work. Our proposed sampling scheme (based on approximating the leverage score) and the theoretical guarantee we made are tailored for the node-level regression task in the training phase. The reason we focused on this specific setting was so that we could take advantage of the theoretical results on leverage score sampling and linear regression in order to obtain theoretical results for GNN training using a subsampled graph. In particular, we are helped by the fact that in the linear GCN setting, the optimal solution for the linear regression is a matrix-vector product which can be expressed as $((AX)^{\top}(AX))^{\top}\cdot (AX)^{\top} \mathbf{y}$, where $A$, $X$, and $\mathbf{y}$ denote the adjacency matrix, data matrix, and the labels respectively.
>
> For graph link prediction and classification tasks (under possibly semi-supervised settings), however, since the optimal solution cannot be expressed as a matrix-vector product (due to non-linearities), our proposed algorithm does not extend to the classification problems in a straightforward way and similarly our theoretical guarantee may not hold anymore. That being said, we agree that both node and edge classification problems are interesting. For future work, we want to study whether our same subsampling process can have any guarantee for a node classification task. For more general tasks,  we believe that we need different sampling strategies other than the leverage score sampling and hence we may need other tools for developing a theoretical guarantee for sampling algorithms.
>
> Q3. Is it possible to use this for link prediction tasks when dropping edges for validation? Is there still any theory guarantee in this case?
>
> That is an interesting question. As mentioned above, our proposed sampling scheme is a node subsampling scheme that selects some representative nodes based on leverage scores and uses all their edges and is tailored for predicting labels (continuous value) of each node. It is not straightforward to apply our algorithm to predicting links between nodes nor to extend the theoretical results to that setting.  However, standard GNN approaches to link prediction still rely on message passing on the graph, requiring the computation of $AX$, and could benefit from a subsampling technique similar to the one we proposed in order to reduce the computational costs. But since link prediction is a fundamentally different inference task, we will need to find an alternative to the leverage score sampling-based approach that we considered. This is an interesting direction for future work.

---

> > ### Comment · Reviewer_J15Y · 2023-08-13
> > **Thank the authors for the effort to address the questions**
> >
> > Thank the authors for the effort to address the questions! But currently, I will hold my score:
> > 1. The assumption is that the graph is dense, which is too strong and weakens its potential application in real cases since most graph structures are sparse. A densely connected graph means that most of the nodes are similar, and maybe the graph neural network is not even useful in such a case, e.g., consider the extreme case when the elements of the adjacency matrix are all-ones.
> > 2. The authors didn't provide the wall time of the sparse multiplication. The primary concern is how dense the graph should be to make the proposed method useful. It is essential to know when is the proposed method faster than the sparse multiplication.
> > 3. The current method only works for limited tasks/structures, further weakening the contribution.

---

> > > ### Author Response · Authors · 2023-08-15
> > > **Additional response to comment 1**
> > >
> > > C1. The assumption is that the graph is dense, which is too strong and maybe the graph neural network is not even useful in such a case, e.g., consider the extreme case when the elements of the adjacency matrix are all-ones.”
> > >
> > > Thank you for the further comments. First, we would like to highlight that the graph we consider is weighted. In other words, the $a_{ij}$ entries of the adjacency matrix are non-negative real values, and not restricted to $\{0,1\}$. This can be used to capture the strength of the connection between nodes $i$ and $j$. Depending on the weights, a dense matrix with different entry values is not necessarily similar to a matrix with all ones and may not correspond to the graph where most of the nodes are similar. An adjacency matrix with non-negative real values could represent, for example, a pairwise similarity matrix and, in such a case, one would expect a fairly dense matrix (even if many of the entries are close to zero but strictly positive).
> > >
> > > In addition, our definition of "dense" is just that the number of non-zero entries scales as $n^2$, but it could still be a small percentage of the edges. For example, an Erdos-Renyi graph with constant edge probability $p$ with have a number of edges that scales as $n^2$.
> > > Lastly, although our theoretical guarantee requires that the adjacency matrix is dense, we would like to highlight that our proposed sampling algorithm works reasonably well empirically, even for sparse graphs. Referring to Figure 2(c) and Figure 3(c) (in the supplementary material), the proposed algorithm outperforms the baselines that we considered for the ego-facebook dataset where the graph is sparse. Hence, we believe that our algorithm may offer benefits in many real-world scenarios.

---

> > > > ### Author Response · Authors · 2023-08-15
> > > > **Additional response to comment 2**
> > > >
> > > > C2. The authors didn't provide the wall time of the sparse multiplication. The primary concern is how dense the graph should be to make the proposed method useful. It is essential to know when is the proposed method faster than the sparse multiplication.
> > > >
> > > > First, we would like to clarify that our proposed method is not meant as an alternative to sparse matrix multiplication for computing $AX$. Our method performs a subsampling on the graph, which can be used to speed the training of a linear GCN, with theoretical accuracy guarantees. Hence, our approach can be used in combination with sparse matrix multiplication. We would like to emphasize two points:
> > > >
> > > > 1. Our proposed sampling algorithm consists of two steps (as illustrated in Figure 1). It first obtains a reduced size of $A$ via approximate leverage score sampling. Then the message passing operation via matrix multiplication of the reduced $A$ and the data matrix is performed. That’s where the sparse matrix multiplication of the reduced $A$ and the data matrix can be combined. The design of the sampling strategy (along with its theoretical guarantee) is one of our core technical contributions and it is independent of the sparse matrix multiplication, which is one specific way to perform the matrix multiplication once the samples are known.
> > > > As per the reviewer’s request, we have run further experiments to compare the wall-clock time of our proposed scheme (adopting sparse matrix multiplication) and that of using exact $AX$ obtained by sparse matrix multiplication. Please see the results below. In particular, we consider the same three datasets previously presented in the attached pdf.
> > > > Notice that the computation time of both our approach and using exact $AX$ become faster due to the adoption of sparse matrix multiplication and the reduction time was larger for using exact $AX$ (especially for the ogbn-arxiv dataset which is sparse). However, we observe that our proposed scheme is faster than the regression by using the exact multiplication of $AX$. Furthermore, for the synthetic dataset where $A$ and $X$ are generated according to a Gaussian distribution and hence $A$ is dense, we observe that the sparse matrix multiplication doesn’t help speed up the computation time and the matrix multiplication with scipy.sparse package took even longer than the matrix multiplication with numpy package.
> > > >
> > > > 2. Our proposed scheme is not only useful for speeding up the training of linear GCNs. It is also useful in partial observation settings where the algorithm only has access to a partial view of the graph, and we provide theoretical guarantees that this partial view is still sufficient to accurately perform the regression task.  On the contrary, sparse matrix multiplication is purely a technique for computational purposes and still requires complete information on $A$ and $X$. For instance, consider a social network setting where, in order to know friendships between users, a learner needs to use an API to query the friendships. Note that one needs fewer queries for using our method (since it requires partial $A$) than for using the sparse matrix multiplication (which needs the queries for obtaining the entire $A$). If the graph is large, the query complexity gain becomes larger.
> > > >
> > > >
> > > > Further experiments with adopting sparse matrix multiplication:
> > > >
> > > > Dataset 1: ogbl-ddi (a dense graph with 4,267 nodes):
> > > >
> > > > Wall-clock time with the use of full AX: 1.0224 (sec)
> > > >
> > > > Wall-clock time with the use of Algorithm1 and leverage score sampling: 0.8411 (sec)
> > > >
> > > > Dataset 2: ogbn-arxiv (a sparse graph with 169,343 nodes):
> > > >
> > > > Wall-clock time with the use of full AX: 6.6904 (sec)
> > > >
> > > > Wall-clock time with the use of Algorithm1 and leverage score sampling: 0.8635 (sec)
> > > >
> > > > Dataset 3: Synthetic dataset (a dense graph where both $A$ and $X$ are generated according to Gaussian distribution with mean 0 and variance 1)
> > > >
> > > > Wall-clock time with the use of full AX: 2452.1392 (sec)
> > > >
> > > > Wall-clock time with the use of Algorithm1 and leverage score sampling: 58.2954 (sec)

---

> > > > > ### Author Response · Authors · 2023-08-15
> > > > > **Additional response to comment 3**
> > > > >
> > > > > C3. The current method only works for limited tasks/structures, further weakening the contribution.
> > > > >
> > > > > In connection to the first answer, we believe that the (dense) graph structures that we consider can be commonly found in real cases since we consider weighted graphs scenarios in general, where weights can be strictly positive but have small values. Besides, we would like to highlight that although our proposed scheme has a theoretical guarantee for dense matrices, it works reasonably well empirically for sparse matrices as well (Please refer to Figure 2(c) and Figure 3(c) in our draft).
> > > > >
> > > > > We also would like to note that regression tasks are one of the most popular learning tasks that try to predict continuous-valued labels. In addition, we have provided some comments and further experiments (please see our response and the attached pdf) regarding how to extend beyond the regression tasks and to the more-than-two-layer case as well as to the non-linear GCNs.

---

> > > > > > ### Comment · Reviewer_J15Y · 2023-08-16
> > > > > >
> > > > > > Thanks to the author for the effort to further address the concerns! Although the authors alleviate my concerns about the limitation of the application and the sparse multiplication, I believe considerable content should be modified/added to the current submission by reading all reviews:
> > > > > > 1. Comparison with other baselines.
> > > > > > 2. The real-world tasks.
> > > > > >
> > > > > > Thus, I will increase my score but lean to reject.

---

> > > > > > > ### Author Response · Authors · 2023-08-21
> > > > > > > **Dear Reviewer J15Y**
> > > > > > >
> > > > > > > We thank the reviewer once again for a helpful discussion on our paper. If we may, we would like to highlight that our original submission introduced a new non-trivial graph subsampling algorithm for the training of linear GCNs, a detailed theoretical analysis based on leverage scores and spectral approximation, and experimental validation of the theoretical claims.
> > > > > > >
> > > > > > > Based on the reviewers' feedback, we will improve the paper by adding (1) comparisons with the sampling strategies utilized in GraphSage and GraphSaint and (2) empirical results on non-linear GCNs, and (3) potential extensions to other real-world tasks. We hope that our responses and discussion can illuminate the potential and significance of our paper.

---

### Official Review · Reviewer_S2HF · 2023-07-07

**Soundness:** 4 excellent
**Presentation:** 3 good
**Contribution:** 2 fair
**Rating:** 3
**Confidence:** 3

**Summary:**

The paper presents an efficient GNN training solution through node subsampling and leverage score sampling, which is proven to be efficient in learning a regression model with bounded entry access and running time.

**Strengths:**

* The proposed technique leads to a proven efficient approach for GNN training, with potentially many real applications.

 * Experimental study on real graph data validates the proposed technique on real world graphs, showing significant improvement over baseline sampling strategies.

**Weaknesses:**

* It is interesting to see the proposed approach for GNN training, as efficiency of GNN on big graphs is a headache for many real world problems. However, although being different approaches, we can find the idea of subsampling used in various GNN models, but it is not well compared in this paper. For example, GraphSage performs neighbor subsampling and GraphSaint subsamples subgraphs from a big graph. The authors should compare the proposed approach with those different sampling approaches to show the pros and cons.

* The experimental results show the significant efficiency of the proposed method. However, since it is for GNN training, the only evaluation on MSE would draw some concerns, if MRS is the only metrics GNN should be concerned about. If not, the authors should expand the experimental study for a comprehensive understanding/evaluation.

* Compared to MSE, it is critical to see how much typical GNN machine learning tasks would be impacted by the subsampling. For example, graph link prediction, node classification, label prediction, etc. This paper does not include evaluation form those perspectives.

* Although the big-O notation of complexity analysis is quite helpful, the experimental study of efficiency should be provided, if the training time is also significantly reduced with comparable performance.

* It is a bit concerned that, for studying GNN efficiency, the authors selected two small graphs from OGB, although much bigger graphs exist.

**Questions:**

N/A

---

> ### Author Rebuttal · Authors · 2023-08-10
>
> W1. Comparison with other baselines (e.g., GraphSage and GraphSaint) and the pros and cons:
>
> We thank the reviewer for the suggestion. Please see the comments in the common response section and the attached pdf for additional experiments.
>
> W2. MSE was the only evaluation criterion:
> Since we focused on a regression task, we believe that the MSE is a natural performance metric. Furthermore, our theoretical guarantee is in terms of the L2 error in the regression task, and the main goal of our experiments was to validate the theoretical guarantees. In future work, once we consider other learning tasks, such as node classification, different performance metrics may be needed.
>
> W3. Applicability of our proposed algorithm to graph link prediction and node classification:
> We thank the reviewer for suggesting interesting directions for future work. Our proposed sampling scheme (based on approximating the leverage score) and the theoretical guarantee we made are tailored for the node-level regression task in the training phase. The reason we focused on this specific setting was so that we could take advantage of the theoretical results on leverage score sampling and linear regression in order to obtain theoretical results for GNN training using a subsampled graph. In particular, we are helped by the fact that in the linear GCN setting, the optimal solution for the linear regression is a matrix-vector product which can be expressed as $((AX)^{\top}(AX))^{\top}\cdot (AX)^{\top} \mathbf{y}$, where $A$, $X$, and $\mathbf{y}$ denote the adjacency matrix, data matrix, and the labels respectively.
>
> For graph link prediction and classification tasks, however, since the optimal solution cannot be expressed as a matrix-vector product (due to non-linearities), our proposed algorithm does not extend to the classification problems in a straightforward way and similarly our theoretical guarantee may not hold anymore. That being said, we agree that both node and edge classification problems are interesting. For future work, we want to study whether our same subsampling process can have any guarantee for a node classification task. For more general tasks,  we believe that we need different sampling strategies other than the leverage score sampling and hence we may need other tools for developing a theoretical guarantee for sampling algorithms.
>
> W4. Lack of run-time comparison:
>
> We thank the reviewer for pointing out this issue. We leave comments in the common response section and provide additional experimental results in the attached pdf.
>
> W5. Experiments on two small graphs:
>
> We would like to first highlight our main contribution lies in providing an efficient sampling algorithm for learning linear GNNs with a large number of nodes with a theoretical guarantee (please see the common response for details). We agree that we include experiments on rather smaller graphs, so we have run further experiments to provide wall-clock runtimes for the experiments and also empirical results on nonlinear GNNs for various sizes of graphs. We refer to the common response above and the attached pdf file for detailed experiment results. We will include additional experiments on large graphs in the main draft if the paper is accepted.

---

> > ### Comment · Reviewer_S2HF · 2023-08-15
> >
> > I would like to thank the response from the authors. The detailed response helps clarify some ambiguity in the original paper and also improves the soundness via the comparison with additional baselines. I will update the scores accordingly.

---

> > > ### Author Response · Authors · 2023-08-15
> > >
> > > We thank the reviewer for the positive feedback and helpful discussion. Based on the discussion and feedback, we will keep polishing the revision to a better version. We are also happy to engage in further discussion if there are any remaining questions.
> > > We look forward to seeing an updated score, as you mentioned.

---

> > > ### Author Response · Authors · 2023-08-18
> > > **Dear Reviewer S2HF**
> > >
> > > Dear Reviewer S2HF, I hope this message finds you well.
> > >
> > > We deeply appreciate your insightful comments on our manuscript and positive feedback on our response. We are also glad that ambiguities are clarified and the soundness of our theoretical result has been improved through our response and further experiments. We are making efforts to integrate our responses and further experimental results to revise our manuscript.
> > >
> > > We genuinely hope that our detailed responses illuminate the potential and significance of our research. If you believe our responses have addressed your concerns, would you kindly consider showing further support by re-evaluating the score?
> > >
> > > This is a friendly reminder that it seems like the score has not been updated yet (although you mentioned to update the score). Such an endorsement would greatly enlarge the chance of this paper being accepted. Warm regards and our deepest gratitude for your time and expertise.

---

### Author Rebuttal · Authors · 2023-08-10

We thank the reviewers for their valuable comments. For weaknesses/questions raised by multiple reviewers, we provide common responses. For the remaining comments, we provide point-by-point responses below.

We would like to first highlight what we see as the main contribution of our paper. Our paper provides an efficient sampling algorithm for learning linear GNNs with a large number of nodes with a theoretical guarantee. Specifically, for a two-layer linear GNN with adjacency matrix $A \in \mathbb{R}^{n \times n}$ and data matrix $X \in \mathbb{R}^{n \times d}$, we show that it is possible to learn the model accurately while observing only $O(nd \log(n))$ entries of $A$. This in turn yields a run-time of $O(nd^2\log n)$, avoiding the computation time $O(n^2d)$ of using the full matrix multiplication $AX$. This reduction in run-time complexity is particularly significant when the number of nodes $n$ is large and the feature dimension is much smaller compared to $n$ (i.e., $d \ll n$).
We would also like to highlight that this result holds even when the graph adjacency matrix and the data matrix are dense in which case sparse matrix multiplication would not help much. Hence the proposed algorithm bears practical importance as well.
While we view our main contribution to be of a theoretical flavor, we agree with several of the reviewers' comments requesting additional experimental validation, which we discuss next.

W1. Lack of run-time comparison:

We note that the experiments in the original submission focused on the tradeoff between the number of observed entries and regression accuracy. In the attached pdf, we provide the computation time comparison for the end-to-end training process for a regression task. In particular, we compared the wall-clock time of performing a regression task with full $AX$ computation and that with Algorithm 1 (our proposed algorithm) that uses partial observations of $A$ and $X$. For a fair comparison, we use the same regression solver and the use the same specification of 48 cores of a x86 64 processor with 503.74GB memory. The results  (see Table 1 in attached pdf) show that our proposed scheme requires orders of magnitude less wall-clock time when the graph is large than the regression with exact $AX$ computation. In particular, for the “ogbn-arxiv dataset” (from Open Graph Benchmark) having 17K nodes and 1M edges,  our algorithm runs about 40x faster than the regression with the exact computation of $AX$.

In addition to run-time, we expect our proposed approach to offer significant improvements in terms of memory usage. As a simple experiment in terms of memory usage, in Table 2 in the attached pdf,  we provide the peak memory usage comparison for the same datasets during the sampling and regression steps of the algorithm. In the best case, the proposed algorithm requires 1414× less memory than the algorithm using the exact computation of $AX$ in the best case.

W2. Comparison with other baselines (e.g., GraphSage and GraphSaint) and the pros and cons:

Thanks for the suggestion. In the original submission, we focused on comparisons with two baselines (full computation of $AX$ and reduced computation via uniform sampling) to validate and highlight the conceptual contributions of our approach. But we agree that comparing our approach to popular methods such as GraphSage and GraphSaint is helpful. The pros and cons between our algorithm and other baselines (GraphSage and GraphSaint) are summarized below:

1. Both our algorithm and the baselines use some form of subsampling of graphs motivated by the large computational/storage complexity of running GNNs with large-scale graphs. However, our algorithm comes with a theoretical guarantee of getting $(1+\epsilon)$-approximation guarantee to the optimal MSE (for the specific setting of a two-layer linear GCN), while only observing $O(nd \log n)$ entries of $A$. GraphSage and GraphSaint, to the best of our knowledge, do not provide similar theoretical guarantees.

2. To obtain the performance guarantee, our algorithm employs a more complicated sampling strategy based on leverage score sampling, which leads to a non-uniform subsampling of the graph. GraphSage uses uniform sampling for choosing the neighborhood of a node when performing a feature aggregation.

Furthermore, we demonstrate comparisons between our sampling approach and the sampling strategies utilized in (1) GraphSage and (2) GraphSaint (not the tools themselves), in addition to applying regression solvers. Figure 2 in the attached file demonstrates the MSE of our proposed algorithm, the baselines we have considered, and GraphSage and GraphSaint. Figures 2(a) and 2(b) demonstrate that the sampling ideas from GraphSage and GraphSaint do not seem to outperform our proposed sampling technique (tailored for the graph-weighted linear regression).

W3. Extension to nonlinear GCN:

We thank reviewers for suggesting interesting directions for future work. As illustrated in Figure 2(d) in the attached pdf, we have run additional experiments to demonstrate our proposed algorithm's efficacy on nonlinear GCNs. Using ReLU activation function and one hidden layer, Figure 2(d) plots the mean squared error as a function of the observation budget, shown as a percentage of the number of observed nodes in the graph. Interestingly, we observe that running the non-linear GCN with our proposed sampling techniques have a similar performance to the one with exact $(AX, \mathbf{y})$, as the budget increases. Also, the error reduction on Algorithm 1 from Regression with uniformly sampled $(AX, \mathbf{y})$ is 60% when the budget is at 5%.
Besides, as compared to Figure 2(c), we observe that although our proposed algorithm is designed to optimize the MSE on linear GCNs, the performance works even better for the non-linear case. It would be interesting to see whether a slight modification of our node subsampling technique will lead us to prove a theoretical guarantee.

---

### Decision · Program_Chairs · 2023-09-21

**Decision:**

Accept (poster)

**Comment:**

This paper proposes an efficient sampling method for linear Graph Neural Networks. Overall, this is a well-written theoretical paper that proposes a new algorithm of node subsampling for GNNs and evaluates the reasonable upper bound of the approximation error. The analyzed model and task are limited to a linear model and a regression for enabling theoretical analysis.

This paper has both positive and negative reviews, and thus this is a borderline paper. There is some limitation (it is a linear model), however, the idea itself is interesting and theoretically well-supported. Moreover, the reviewer who agrees to accept is excited about the paper. One of the weaknesses of this paper is that it does not compare with other nonlinear GNN models, as pointed out by a reviewer. If this paper proposes a nonlinear GNN method, this could be a reason for rejection. However, this paper considers a linear setting; it sounds OK not to compare with nonlinear models.

Based on the above reasons, I vote for acceptance.